# Q-learning with temporal memory to navigate turbulence

**Marco Rando[1]\*, Martin James[2], Alessandro Verri[1], Lorenzo Rosasco[1], Agnese Seminara[2]\***

[1]MaLGa, Department of Computer Science, Bioengineering, Robotics and Systems Engineering, University of Genova, Genoa, Italy; [2]MaLGa, Department of Civil, Chemical and Environmental Engineering, University of Genoa, Genova, Italy

## eLife Assessment

This **important** study uses reinforcement learning to study how turbulent odor stimuli should be processed to yield successful navigation. The authors find that there is an optimal memory length over which an agent should ignore blanks in the odor to discriminate whether the agent is still inside the plume or outside of it, complementing recent studies using recurrent neural networks and finite state controllers to identify optimal strategies for navigating a turbulent plume. The strength of evidence is **compelling**, presenting a novel approach to understanding optimal representations for navigation in stochastic sensory environments.

**\*For correspondence:**
marco.rando@edu.unige.it (MR);
agnese.seminara@unige.it (AS)

**Abstract** We consider the problem of olfactory searches in a turbulent environment. We focus on agents that respond solely to odor stimuli, with no access to spatial perception nor prior information about the odor. We ask whether navigation to a target can be learned robustly within a sequential decision making framework. We develop a reinforcement learning algorithm using a small set of interpretable olfactory states and train it with realistic turbulent odor cues. By introducing a temporal memory, we demonstrate that two salient features of odor traces, discretized in a few olfactory states, are sufficient to learn navigation in a realistic odor plume. Performance is dictated by the sparse nature of turbulent odors. An optimal memory exists which ignores blanks within the plume and activates a recovery strategy outside the plume. We obtain the best performance by letting agents learn their recovery strategy and show that it is mostly casting cross wind, similar to behavior observed in flying insects. The optimal strategy is robust to substantial changes in the odor plumes, suggesting minor parameter tuning may be sufficient to adapt to different environments.

## Introduction

Bacterial cells localize the source of an attractive chemical even if they hold no spatial perception. They respond solely to temporal changes in chemical concentration, and the result of their response is that they move toward attractive stimuli by climbing concentration gradients (**Berg, 1975**). Larger organisms also routinely sense chemicals in their environment to localize or escape targets, but cannot follow chemical gradients since turbulence breaks odors into sparse pockets and gradients lose information (**Murlis et al., 1992**; **Vergassola et al., 2007**; **Shraiman and Siggia, 2000**; **Balkovsky and Shraiman, 2002**; **Reddy et al., 2022**). The question of which features of turbulent odor traces are used by animals for navigation is natural, but not well understood. Beyond olfaction, some animals could also use prior spatial information to navigate (**Cardé, 2021**; **Schal, 1982**; **Gire et al., 2016**; **Baker et al., 2018**), but if and how chemosensation and spatial perception are coupled is still not clear.

**eLife digest** Many animals use odors to locate mates, food, and to avoid danger. Unlike light, which travels in straight lines, odors are carried by turbulent air or water, leading to intermittent whiffs separated by long gaps with no detectable scent. These patchy odor landscapes can make it difficult for animals to decide which direction to move in.

Despite these challenges, animals are remarkably good at using odors to navigate. While previous studies have modelled this behavior computationally, the most principled models often relied on complex concepts of memory, that were not directly interpretable. In particular, what must be remembered about past odor detections and for how long remained unclear.

To investigate this, Rando et al. developed an algorithm that enables agents to learn to navigate by trial and error, responding only to a short excerpt of past odor detections. Agents had no prior knowledge about the odor nor access to spatial information, other than their ability to orient relative to the wind. The simulated environment mimicked realistic odor plumes in turbulent air and the algorithm was given a short-term memory to track changes in a limited set of specific odor-related signals over time.

Analysis showed that there is an optimal length of memory that helps the agent ignore temporary gaps in the odor signal while still recognizing when it has fully exited the plume. This allowed the agent to activate a strategy to return to the scent plume only when truly necessary. When it was allowed to learn behavior both within and outside the plume, it performed better than when using fixed strategies based on animal behavior. Interestingly, the learned strategy often resembled the casting behavior, seen in flying insects, which involves a side-to-side search in the crosswind direction to relocate odor plumes.

Overall, the work of Rando et al. shows that simple odor signals and a basic form of temporal memory are enough to learn effective navigation in turbulent environments with no prior knowledge of the odor environment. The algorithm performed reliably, reaching the odor source in 90% to 100% of trials. These findings help explain how animals might use short-memory of odor to navigate in space, even in unknown or variable environments and could be used to develop search algorithms for robots in complex real-world settings like disaster zones or polluted areas.

An algorithmic perspective to olfactory navigation in turbulence can shed light on some of these questions. Without aiming at an exhaustive taxonomy, see for example *Celani and Panizon, 2024* for a recent review, we recall some approaches relevant to put our contribution in context. One class of methods is biomimetic algorithms, where explicit navigation rules are crafted taking inspiration from animal behavior. An advantage of these methods is interpretability, in the sense that they provide insights into features that effectively achieve turbulent navigation, for example: odor presence/absence (*Baker, 1990*; *Kramer, 1997*; *Belanger and Willis, 1988*; *Balkovsky and Shraiman, 2002*); odor slope at onset of detection (*Atema, 1996*; number of detections in a given interval of time (*Michaelis et al., 2020*) and the time of odor onset (*Demir et al., 2020*). On the flip side, in biomimetic algorithms, behaviors are hardwired and typically reactive, not relying on any optimality criterion.

A way to tackle this shortcoming is to cast olfactory navigation within a sequential decision-making framework (*Sutton and Barto, 1998*). In this context, navigation is formalized as a task with a reward for success; by maximizing reward, optimal strategies can be sought to efficiently reach the target. A byproduct is that most algorithmic choices can often be done in a principled way. Within this framework, some approaches make explicit use of spatial information. Bayesian algorithms use a spatial map to guess the target location and use odor to refine this guess or 'belief'. A prominent algorithm for olfactory navigation based on the concept of belief is the information-seeking algorithm (*Vergassola et al., 2007*) akin to exploration heuristics widely used in robotics (*Cassandra et al., 1996*; *LaValle, 2006*; see e.g. *Loisy and Eloy, 2022*; *Ishida et al., 2012*). Using Bayesian sequential decision making and the notion of beliefs, navigation can be formalized as a Partially Observable Markov Decision Process (POMDP; *Krishnamurthy, 2016*; *Hauskrecht, 2000*; *Shani et al., 2013*), that can be approximatively solved (*Rigolli et al., 2022b*; *Heinonen et al., 2023*; *Loisy and Heinonen, 2023*). POMDP approaches are appealing since beliefs are a sufficient statistic for the entire history of odor

detections. However, they are computationally cumbersome. Further, they leave the question open of whether navigation as sequential decision making can be performed using solely olfactory information.

Recently, two algorithms studied navigation as a response to olfactory input alone (*Singh et al., 2023*; *Verano et al., 2023*). In *Singh et al., 2023*, artificial neural networks were shown to learn near-optimal strategies as a response to odor and instantaneous flow direction, although they were trained on odor cues with limited sparsity, and training with sparse odor cues typical of turbulence remains to be tested. In *Verano et al., 2023*, an approach based on finite state controllers was proposed. Here, optimization was done assuming fixed known mean flow direction and using a model-based technique, relying on prior knowledge of the likelihood to detect the odor in space, hence still using spatial information. A different model-free optimization could also be considered, avoiding spatial information, but this latter approach also remains to be tested. More generally, all the above approaches manipulate internally the previous history (memory) of odor detections. In this sense, they are less interpretable, since the features of odor traces that drive navigation do not emerge explicitly.

In this paper, we propose a reinforcement learning (RL) approach to navigation in turbulence based on a set of interpretable olfactory features, with no spatial information other than the ability to orient relative to the mean flow, and highlight the role played by memory within this context. More precisely, we learn optimal strategies from data by training tabular Q learning (*Sutton and Barto, 1998*) with realistic odor cues obtained from state-of-the-art Direct Numerical Simulations of turbulence. From the odor cues, we define features as moving averages of odor intensity and sparsity: the moving window is the temporal memory and naturally connects to the physics of turbulent odors. States are then obtained by discretizing such features. Due to sparsity, agents may detect no odor within the moving window. We show there is an optimal memory minimizing the occurrence of this 'void state'. The optimal memory scales with the blank time dictated by turbulence as it emerges from a trade-off requiring that: (*i*) short blanks – typical of turbulent plumes – are ignored by responding to detections further in the past and (*ii*) long blanks promptly trigger a recovery strategy to make contact with the plume again. We leverage these observations to tune the memory adaptively, by setting it equal to the previous blank experienced along an agent's path. With this choice, the algorithm tests successfully in distinct environments, suggesting that tuning can be made robustly to enable generalization. The agent learns to surge upwind in most non-void states and to recover by casting crosswind in the absence of detections. Optimal agents limit encounters with the void state to a narrow band right at the edge of the plume. This suggests that the temporal odor features we considered effectively

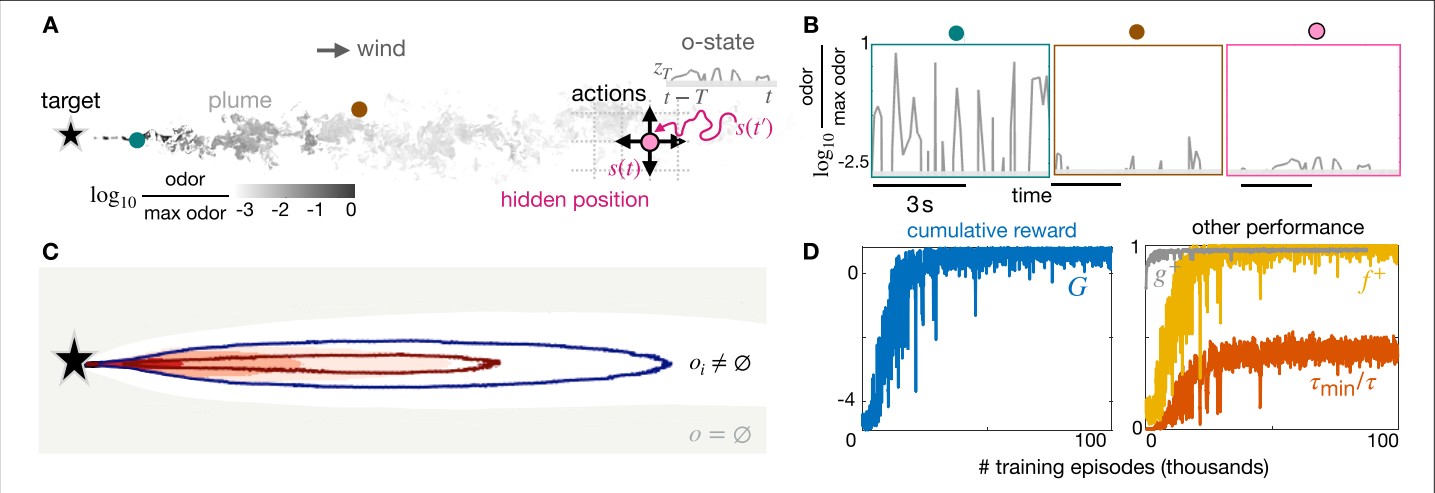

**Figure 1.** Learning a stimulus-response strategy for turbulent navigation. (**A**) Representation of the search problem with turbulent odor cues obtained from Direct Numerical Simulations of fluid turbulence (gray scale, odor snapshot from the simulations). The discrete position $s$ is hidden; the odor concentration $z_T = z(s(t'), t')|t - T \leq t' \leq t$ is observed along the trajectory $s(t')$, where $T$ is the sensing memory. (**B**) Odor traces from direct numerical simulations at different (fixed) points within the plume. Odor is noisy and sparse, information about the source is hidden in the temporal dynamics. (**C**) Contour maps of olfactory states with nearly infinite memory ($T = 2598$): on average, olfactory states map to different locations within the plume, and the void state is outside the plume. Intermittency is discretized in three bins defined by two thresholds: 66% (red line) and 33% (blue line). Intensity is discretized in 5 bins (dark red shade to white shade) defined by four thresholds (percentiles 99%, 80%, 50%, 25%). (**D**) Performance of stimulus-response strategies obtained during training, averaged over 500 episodes. We train using realistic turbulent data with memory $T = 20$ and backtracking recovery.

predict when the agent is exiting the plume and point to an intimate connection between temporal predictions and spatial navigation.

## Results

### Background

Given a source of odor placed in an unknown position of a two-dimensional space, we consider the problem of learning to reach the source, *Figure 1A*. We formulate the problem as a discrete Markov Decision Process by discretizing space in tiles, also called 'gridworld' in the reinforcement learning literature (*Sutton and Barto, 1998*).

In this problem, an agent is in a given state $s$ which is one of a discrete set of $n$ tiles: $s \in S := \{s_1, ..., s_n\}$. At each time step, it chooses an action $a$ which is a step in any of the coordinate directions $a \in \{$upwind, downwind, crosswind-left, crosswind-right$\}$. Directions are labeled relative to the mean wind, which is assumed known. In our figures, the flow always goes from left to right, hence the actions upwind, downwind, crosswind-right, and crosswind-left correspond in the figures to a step left, right, up, and down, respectively. The goal is to find sequences of actions that lead to the source as fast as possible and is formalized with the notion of policy and reward, which we will introduce later. If agents have perfect knowledge of their own location and of the location of the source in space, the problem reduces to finding the shortest path.

### Using time *vs* space to address partial observability

In our problem, however, the agent does not know where the source is; hence, its position $s$ relative to the source is unknown or 'partially observed'. Instead, it can sense odor released by the target. In the language of RL, odor is an 'observation' – but does it hold information about the position $s$? The answer is yes: several properties of odor stimuli depend on the distance from the source (*Boie et al., 2018*; *Ackels et al., 2021*; *Nag and van Breugel, 2024*). However, in the presence of turbulence, information lies in the statistics of the odor stimulus. Indeed, when odor is carried by a turbulent flow, it develops into a dramatically stochastic plume, that is a complex and convoluted region of space where the fluid is rich in odor molecules. Turbulent plumes break into structures that distort and expand while they travel away from their source and become more and more diluted (*Falkovich et al., 2001*; *Shraiman and Siggia, 2000*; *Celani et al., 2014*; *Reddy et al., 2022*), see *Figure 1A*. As a consequence, an agent within the plume experiences intermittent odor traces that endlessly switch on (whiff) and off (blank) *Figure 1B*. The intensity of odor whiffs and how they are interleaved with blanks depends on distance from release, as dictated by physics (*Celani et al., 2014*). Thus, the upshot of turbulent transport is that the statistical properties of odor traces depend intricately on the position of the agent relative to the source. In other words, information about the state $s$ is hidden within the observed odor traces.

This positional information can be leveraged with a Bayesian approach that relies on guessing $s$, that is defining the probability distribution of the position, also called belief. This is the approach that has been more commonly adopted in the literature until now (*Rigolli et al., 2022b*; *Heinonen et al., 2023*; *Loisy and Heinonen, 2023*). Note that because of the complexity of these algorithms, only relatively simple measures of the odor are computationally feasible, for example instantaneous presence/absence. Here, we take a different model-free and map-free approach. Instead of guessing the current state $s$, we ignore the spatial position and respond directly to the temporal traces of the odor cues. Two other algorithms have been proposed to solve partial observability by responding solely to odor traces with recurrent neural networks (*Singh et al., 2023*) and finite state controllers (*Verano et al., 2023*) that manipulate implicitly the odor traces. Here, instead, we manipulate odor traces explicitly, by defining memory as a moving window and by crafting a small number of features of odor traces.

### Features of odor cues: definition of discrete olfactory states and sensing memory

To learn a response to odor traces, we set out to craft a finite set of *olfactory states*, $o \in O$, so that they bear information about the location $s$. Defining the olfactory states is a challenge due to the dramatic fluctuations and irregularity of turbulent odor traces. To construct a fully interpretable low-dimensional

state space, we aim at a small number of olfactory states that bear robust information about $s$, that is for all values of $s$. We previously found that pairing features of sparsity as well as intensity of turbulent odor traces predicts robustly the location of the source for all $s$ (**Rigolli et al., 2022a**). Guided by these results, we use these two features extracted from the temporal history of odor detections to define a small set of olfactory states.

We proceed to define a function that takes as input the history of odor detections along an agent's path and returns its current olfactory state. We indicate with $s(t)$ the (unknown) path of an agent, and with $z(s(t), t)$ the observations that is odor concentration along its path. First, we define a sensing memory $T$ and we consider a short excerpt of the history of odor detections $z_T$ of duration $T$ prior to the current time $t$. Formally, $z_T(t) := \{z(s(t'), t') \mid t - T \leq t' \leq t\}$. Second, we measure the average intensity $c$ (moving average of odor intensity over the time window $T$, conditioned to times when odor is above threshold), and intermittency $i$ (the fraction of time the odor is above threshold during the sensing window $T$). Both features $c$ and $i$ are described by continuous, positive real numbers. Third, we define 15 olfactory states by discretizing $i$ and $c$ in 3 and 5 bins, respectively. Intermittency $i$ is bounded between 0 and 1, and we discretize it in 3 bins by defining two thresholds (33% and 66%). The average concentration, $c$, is bounded between 0 and the odor concentration at the source, hence prior information on the source is needed to discretize $c$ using set thresholds. To avoid relying on prior information, we define thresholds of intensity as percentiles, based on a histogram that is populated online, along each agent's path (see Materials and methods). The special case where no odor is detected over $T$ deserves attention, hence we include it as an additional state named 'void state' and indicate it with $o \equiv \emptyset$. When $T$ is sufficiently long, the resulting olfactory states map to different spatial locations (**Figure 1C**, with $T$ equal to the simulation time). Hence, this definition of olfactory states can potentially mitigate the problem of partial observability using temporal traces, rather than spatial maps. But will these olfactory states with finite memory $T$ guide agents to the source?

## Q learning: a map-less and model-free navigation to odor sources

To answer this question, we trained tabular episodic Q learning (**Sutton and Barto, 1998**). In a nutshell, we use a simulator to place an agent at a random location in space at the beginning of each episode. The agent is not aware of its location in space, but it senses odor provided by the fluid dynamics simulator and thus can compute its olfactory state $o$, based on odor detected along its path in the previous $T$ sensing window. It then makes a move according to a set policy of actions $a \sim \pi_0(a|o)$. After the move, the simulator displaces the agent to its new location and relays the agent a penalty $R = -\sigma$ with $\sigma = 0.001$ if it is not at the source and a reward $R = 1$ if it reaches the source. The goal of RL is to find a policy of actions that maximizes the expected cumulative future reward $G = E_\pi(\sum_{t=0}^{\infty} \gamma^t R_{t+1})$ where the expectation is over the ensemble of trajectories and rewards generated by the policy from any initial condition. Because reward is only positive at the source, the optimal policy is the one that reaches the source as fast as possible. To further encourage the agent to reach the source quickly, we introduce a discount factor $\gamma < 1$.

Episodes where the agent does not reach the source are ended after $H_{max} = 5000$ with no positive reward. As it tries actions and receives rewards, the agent learns how good the actions are. This is accomplished by estimating the quality matrix $Q(o, a)$, that is the maximum expected cumulative reward conditioned to being in $o$ and choosing action $a$ at the present time: $Q(o, a) = \max_\pi E_\pi(\sum_{t=0}^{\infty} \gamma^t R_{t+1} | o_t = o, a_t = a)$. At each step, the agent improves its policy by choosing more frequently putatively good actions. Once the agent has a good approximation of the quality matrix, the optimal policy corresponds to the simple readout: $\pi^*(a|o) = \delta(a - a^*(o))$ where $a^*(o) = \arg\max_a Q(o, a)$, for non-void states $o \neq \emptyset$.

### Recovery strategy

To fully describe the behavior of our Q-learning agents, we have to prescribe their policy from the void state $o \equiv \emptyset$. This is problematic because turbulent plumes are full of holes, thus the void state can occur anywhere both within and outside the plume, **Figure 1A**. As a consequence, the optimal action $a^*(\emptyset)$ from the void state is ill-defined. We address this issue by using a separate policy called 'recovery strategy'. Inspired by path integration as defined in biology (**Etienne and Jeffery, 2004**; **Etienne et al., 1996**; **Heinze et al., 2018**), we propose the backtracking strategy consisting of retracing the last $T_a$ steps after the agent lost track of the odor. If at the end of backtracking the agent is still in

the void state, it activates Brownian motion. Backtracking requires that we introduce memory of the past $T_a$ actions. This timescale $T_a$ for activating recovery is conceptually distinct from the duration of the sensing memory – however, here we set $T_a = T$ for simplicity. Backtracking was observed in ants displaced in unfamiliar environments (**Wystrach et al., 2013**), tsetse flies executing reverse turns bringing them back towards the location where they last detected odor (**Torr, 1988**; **Gibson and Brady, 1985**) and cockroaches retracing their steps downwind, sometimes walking all the way back to the release point upon plume loss **Willis et al., 2008**; it was also previously used in computational models (**Park et al., 2016**).

We find that Q-learning agents successfully learn to navigate to the odor source by responding solely to their olfactory state, with no sense of space nor models of the odor cues. Learning can be quantified by monitoring the cumulative reward which continuously improves with further training episodes (**Figure 1D**, left). Improved reward corresponds to agents learning how to reach the source more quickly and reliably with training. Indeed, it is easy to show that the expected cumulative reward $G = \langle e^{-\lambda\tau} - \sigma(1 - e^{-\lambda\tau})/(1 - \gamma) \rangle$, where $\tau$ is a random variable corresponding to time to reach the source and $\gamma = e^{-\lambda\Delta t}$ is the discount factor, with the time step $\Delta t = 1$ (see Materials and methods). Large rewards arise when (*i*) a large fraction $f^+$ of agents successfully reaches the source and (*ii*) the agents reach the source quickly, which maximizes $g^+ = \langle e^{-\lambda\tau}|\text{success} \rangle$. Indeed $G = f^+ G^+ + (1 - f^+)G^-$, where $G^+ = g^+ - \sigma(1 - g^+)/(1 - \gamma)$ and $G^- = -\sigma(1 - e^{-\lambda H_{\max}})/(1 - \gamma)$ is the horizon of the agent that is the maximum time the agent is allowed to search, and after which the search is considered failed. Note that agents starting closer to the target receive larger rewards purely because of their initial position. To monitor performance independently on the starting location, we introduce the inverse time to reach the source relative to the shortest-path time from the same initial location, which goes from 0 for failing agents to 1 for ideal agents $\langle \tau_{\min}/\tau \rangle$, independently on their starting location. Note that this is not the quantity that is optimized for. One may specifically target this performance metrics, which is agnostic on the duration of an agent's path, by discounting proportionally to $t/\tau_{\min}$.

All four measures of performance plateau to a maximum, suggesting learning has achieved a nearly optimal policy (**Figure 1D**). Once training is completed, we simulate the trajectory of test agents starting from any of the about 43,000 admissible locations within the plume and moving according to the optimal policy. Admissible locations are defined as any location where the odor is non-zero at least once within the entire simulation. We will recapitulate performance with the cumulative reward $G$ averaged over the test agents and dissect it into speed $g^+$, convergence $f^+$ and relative time $\langle \tau_{\min}/\tau \rangle$.

## Optimal memory

By repeating training using different values of $T$, we find that performance depends on memory and an optimal memory $T^*$ exists (**Figure 2A**). Why is there an optimal memory? The shortest memory $T = 1$ corresponds to instantaneous olfactory states: the instantaneous contour maps of the olfactory states are convoluted, and the void state is pervasive (**Figure 2C**, top). As a consequence, agents often activate recovery even when they are within the plume. The policy almost always leads to the source ($f^+ = 79\% \pm 13\%$) but follows lengthy convoluted paths ($\tau_{\min}/\tau = 0.14 \pm 0.05$, **Figure 2C**, bottom). As memory increases, the olfactory states become smoother and agents encounter fewer voids (**Figure 2C**, center), perform straighter trajectories ($\tau_{\min}/\tau = 0.5 \pm 0.3$), and reach the source reliably ($f^+ = 95\% \pm 8\%$), **Figure 2A**, bottom. Further increasing memory leads to even less voids within the plume and even smoother olfactory states. However – perhaps surprisingly – performance does not further improve but slightly decreases (at $T = 50$, $f^+ = 94\% \pm 8\%$ and $\tau_{\min}/\tau = 0.38 \pm 0.36$). A long memory is deleterious because it delays recovery from accidentally exiting the plume, thus increasing the number of voids *outside* of the plume (**Figure 2C**, bottom). Indeed, agents often leave the plume accidentally as they measure their olfactory state *while they move*. They receive no warning, but realize their mistake after $T$ steps, when they enter the void state and activate recovery to re-enter the plume. The delay is linear with memory when agents recover by backtracking, but it depends on the recovery strategy (see Materials and methods and **Figure 2—figure supplement 1**).

Thus, short memories increase time in void *within* the plume, whereas long memories increase time in void *outside* the plume: the optimal memory minimizes the overall chances to experience the void (**Figure 2B**). Intuitively, $T^*$ should match the typical duration $\tau_b$ of blanks encountered within the plume, so that voids within the plume are effectively ignored without delaying recovery unnecessarily.

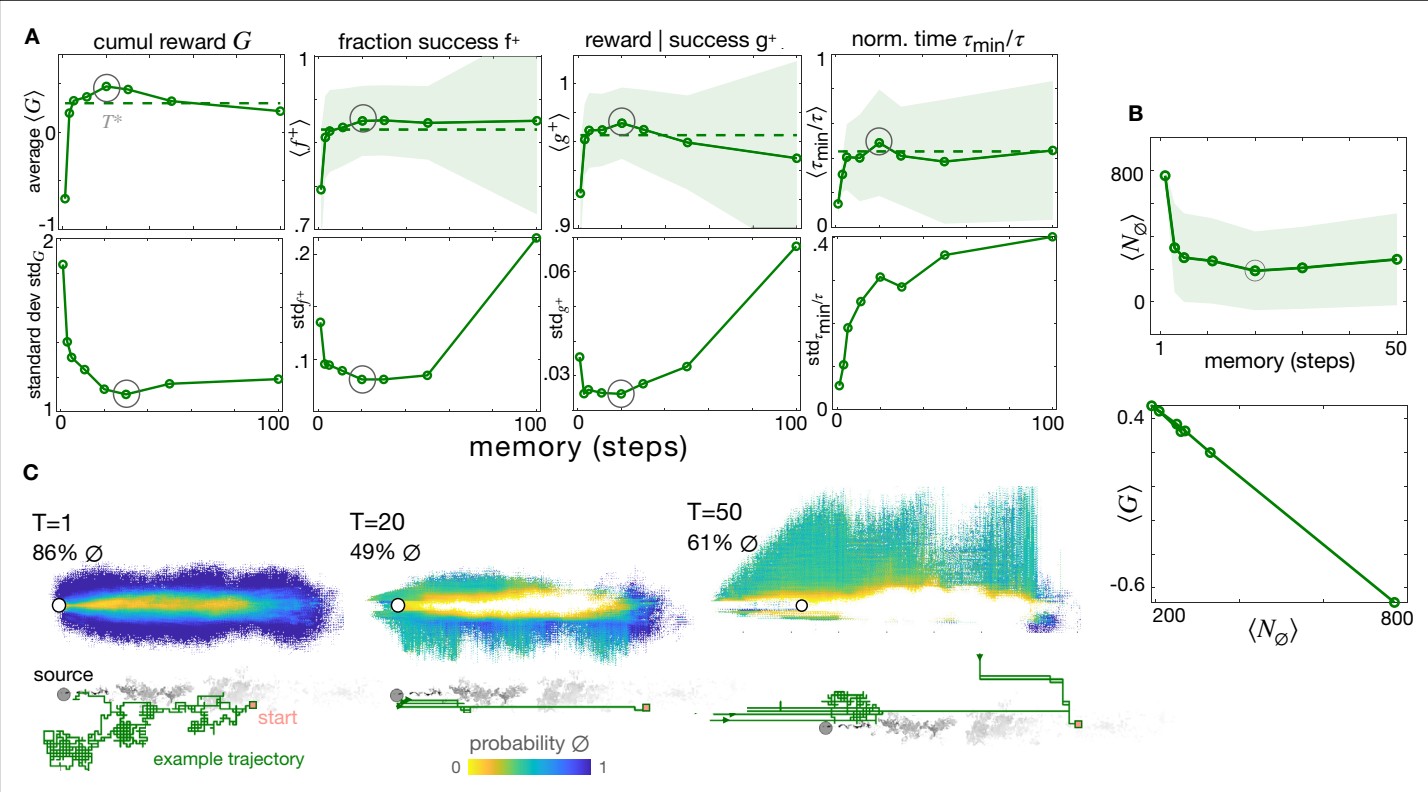

**Figure 2.** The optimal memory $T^*$. (**A**) Four measures of performance as a function of memory with backtracking recovery (solid line) show that the optimal memory $T^* = 20$ maximizes average performance and minimizes standard deviation, except for the normalized time. Top: Averages computed over 10 realizations of test trajectories starting from 43,000 initial positions (dash: results with adaptive memory). Bottom: standard deviation of the mean performance metrics for each initial condition (see Materials and methods). (**B**) Average number of times agents encounter the void state along their path, $\langle N_\emptyset \rangle$, as a function of memory (top); cumulative average reward $\langle G \rangle$ is inversely correlated to $\langle N_\emptyset \rangle$ (bottom), hence the optimal memory minimizes encounters with the void. (**C**) Colormaps: Probability that agents at different spatial locations are in the void state at any point in time, starting the search from anywhere in the plume and representative trajectory of a successful searcher (green solid line) with memory $T = 1$, $T = 20$, $T = 50$ (left to right). At the optimal memory, agents in the void state are concentrated near the edge of the plume. Agents with shorter memories encounter voids throughout the plume; agents with longer memories encounter more voids outside of the plume as they delay recovery. In all panels, shades are ± standard deviation.

The online version of this article includes the following figure supplement(s) for figure 2:

**Figure supplement 1.** The role of temporal memory with Brownian recovery strategy (same as main *Figure 2A*).

Consistently, $\langle \tau_b \rangle$ averaged across all locations and times within the plume is $\langle \tau_b \rangle = 9.97 \pm 41.16$, comparable with the optimal memory $T^*$ (*Figure 2A*).

## Adaptive memory

There is no way to select the optimal memory $T^*$ without comparing several agents or relying on prior information on the blank durations. In order to avoid prior information, we venture to define memory adaptively along each agent's path, using the intuition outlined above. We define a buffer memory $T_b$, and let the agent respond to a sensing window $T < T_b$. Ideally, we would like to set $T \sim \langle \tau_b \rangle$. With this choice, blanks shorter than the average blank are ignored, as they are expected within the plume, whereas blanks longer than average initiate recovery, as they signal that the agent exited the plume. However, agents do not have access to $\langle \tau_b \rangle$ hence we set $T = \tau_b^-$, where $\tau_b^-$ is the most recent blank experienced by the agent. With this choice, the sensing memory $T$ fluctuates considerably along an agent's path, due to turbulence (*Celani et al., 2014* and *Figure 3A–B*). Note that blanks are estimated along paths, thus the statistics of $T$ only qualitatively matches the Eulerian statistics of $\tau_b$. Despite the fluctuations, performance using the adaptive memory nears performance with the optimal memory (*Figure 3C*). This result confirms our intuition that memory should match the blank time. The advantage of adaptive memory is that it relies solely on experience, with no prior information whatsoever.

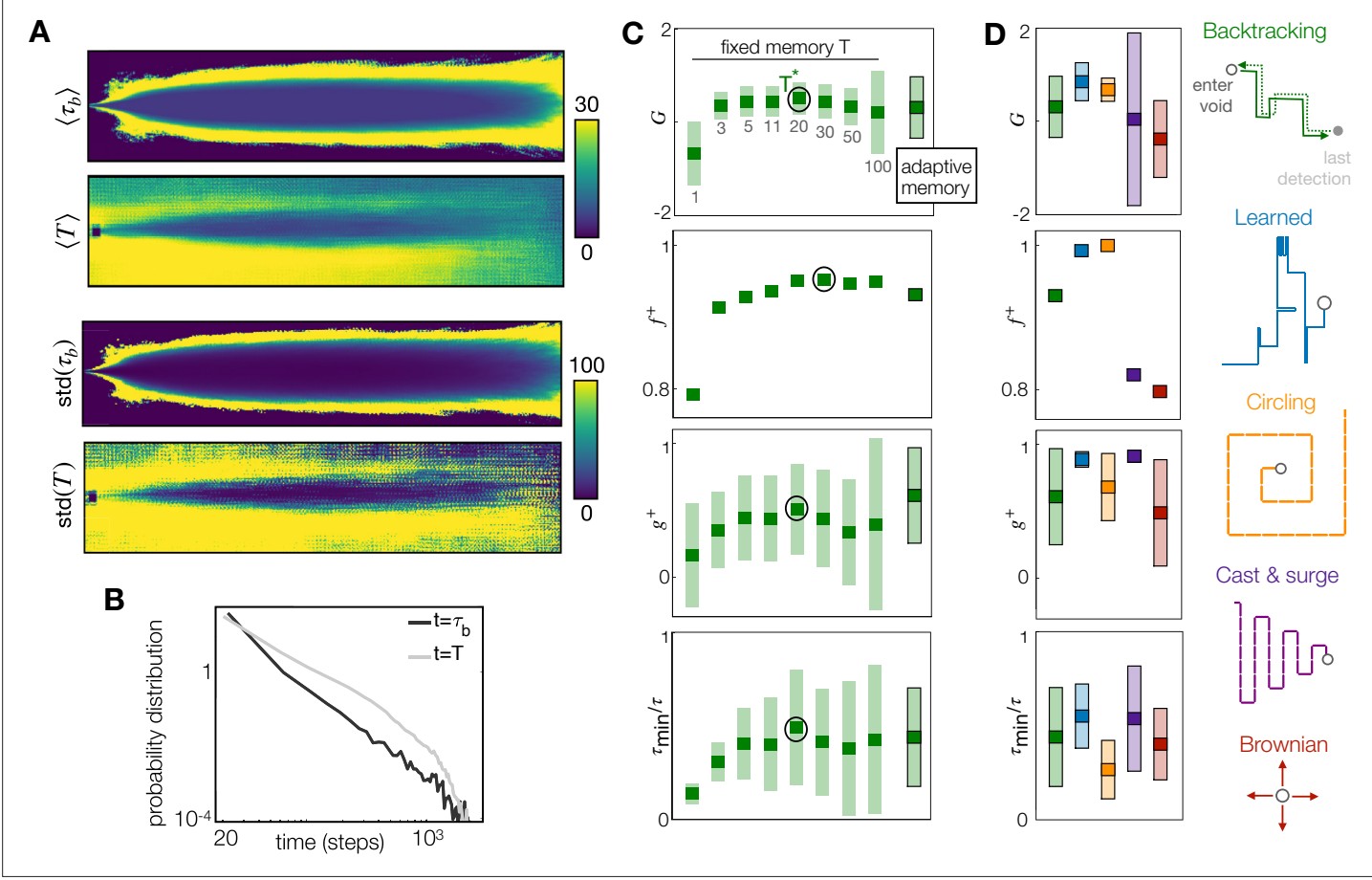

**Figure 3.** The adaptive memory approximates the duration of the blank dictated by physics, and it is an efficient heuristic, especially when coupled with a learned recovery strategy. (**A**) Top to bottom: Colormaps of the Eulerian average blank time $\tau_b$; average sensing memory $T$; standard deviation of Eulerian blank time and of sensing memory. The sensing memory statistics are computed over all agents that are located at each discrete cell, at any point in time. (**B**) Probability distribution of $\tau_b$ across all spatial locations and times (black) and of $T$ across all agents at all times (gray). (**C**) Performance with the adaptive memory nears performance of the optimal fixed memory, here shown for backtracking; similar results apply to the Brownian recovery (*Figure 3—figure supplement 1*). (**D**) Comparison of five recovery strategies with adaptive memory: The learned recovery with adaptive memory outperforms all fixed and adaptive memory agents. In (**C**) and (**D**), dark squares mark the mean, and light rectangles mark ± standard deviation. $f^+$ is defined as the fraction of agents that reach the target at test, hence has no standard deviation.

The online version of this article includes the following figure supplement(s) for figure 3:

**Figure supplement 1.** All four measures of performance across agents with fixed memory and Backtracking vs Brownian recovery (green and red respectively, unframed boxes) and with adaptive memory for Backtracking, Brownian, and Learned recovery (green, red, and blue respectively, framed boxes).

This is unlike $T^*$ which can only be selected using prior information, with no guarantee of generalization to other plumes.

## Learning to recover

So far, our agents combine a learned policy from non-void states to a heuristic from the void state, which we called the recovery strategy. We have considered biologically inspired heuristics where searchers make it back to locations within the plume by retracing their path backward. To further strip the algorithm of heuristics, we ask whether the recovery strategy may be learned, rather than fixed a priori. To this end, we split the void state in many states, labeled with the time elapsed since first entering the void. We pick 50 void states, as less than 50 void states results in no convergence, and states above 50 are useless because they are rarely visited. The counter is reset to 0 whenever the searcher detects the odor. The definition of the 15 non-void states $o_1, \ldots, o_{15}$ remains unaltered. Interestingly, with this added degree of freedom, the agent learns an even better recovery strategy

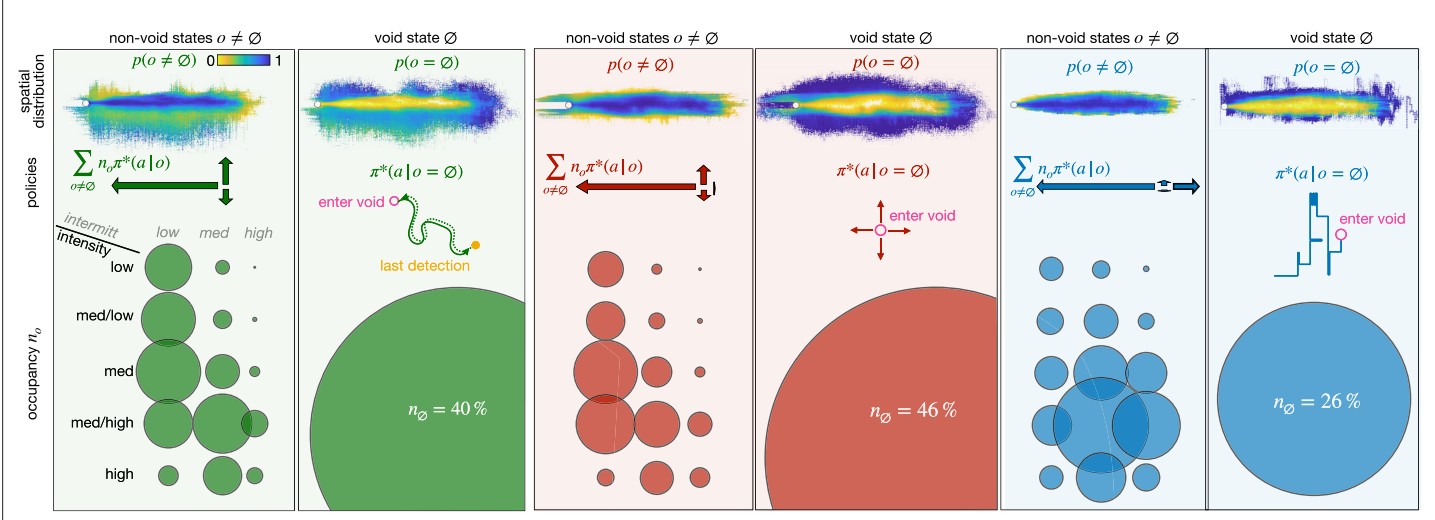

**Figure 4.** Optimal policies with adaptive memory for different recovery strategies: backtracking (green), Brownian (red), and learned (blue). For each recovery, we show the spatial distribution of the olfactory states (top); the policy (center) and the state occupancy (bottom) for non-void states (left) vs the void state $\pi^*(a|\emptyset)$(right). Spatial distribution: probability that an agent at a given position is in any non-void olfactory state (left) or in the void state (right), color-coded from yellow to blue. Policy: actions learned in the non-void states $\sum_{o\neq\emptyset} n_o\pi^*(a|o)$, weighted on their occupancy $n_o$ (left, arrows proportional to the frequency of the corresponding action) and schematic view of recovery policy in the void state (right). State occupancy: fraction of agents that is in any of the 15 non-void states (left) or in the void state (right) at any point in space and time. Occupancy is proportional to the radius of the corresponding circle. The position of the circle identifies the olfactory state (rows and columns indicate the discrete intensity and intermittency respectively). All statistics are computed over 43,000 trajectories, starting from any location within the plume.

The online version of this article includes the following figure supplement(s) for figure 4:

**Figure supplement 1.** Optimal policies for different recovery strategies and adaptive memory.

**Figure supplement 2.** The learned recovery resembles the cast and surge observed in animals, with an initial surge of 5±2 steps and a subsequent motion crosswind starting from either side of the centerline and overshooting to the other side.

as reflected by all our measures of performance (*Figure 3D*). Note that the learned recovery strategy resembles the casting behavior observed in flying insects (*David et al., 1983*), as discussed below. In fact, insects deploy a range of recovery strategies depending on locomotor mode and environment. To corroborate these results, we compare performance using two additional biologically-inspired recovery strategies, i.e. circling (observed in windless environments *Stupski and van Breugel, 2024a*), and cast & surge (*David et al., 1983*) as well as a Brownian recovery which does not have a direct biological relevance but represents a simple computational benchmark. The learned recovery outperforms all heuristic recoveries, as seen by the cumulative reward $G$ (*Figure 3D*). Circling is the second-best recovery and shortly follows the learned recovery. Circling achieves nearly optimal performance by further decreasing failures (metrics $f^+$), but slowing down (metrics $g^+$ and $\tau_{min}/\tau$).

## Characterization of the optimal policies

To understand how different recoveries affect the agent's behavior, we characterize the optimal policies obtained using the three recovery strategies. We visualize the probability of encountering each of the 16 olfactory states, or occupancy (circles in *Figure 4*), and the spatial distribution of the olfactory states.

In the void state, the agent activates the recovery strategy. Recovery from the void state affects non-void olfactory states as well: their occupancy, their spatial distribution, and the action they elicit (*Figure 4*, *Figure 4—figure supplement 1*). This is because the agent computes its olfactory state online, according to its prior history which is affected by encounters with the void state. However, for all recoveries, non-void states are mostly encountered within the plume and largely elicit upwind motion (*Figure 4*, top, center). Thus macroscopically, all agents learn to surge upwind when they detect any odor within their memory, and to recover when their memory is empty. This suggests a considerable level of redundancy which may be leveraged to reduce the number of olfactory states, thus the computational cost. Reducing the number of non-empty olfactory states drastically to just 1

**Table 1.** Parameters of the learned recovery, statistics over 20 independent trainings.

| Initial surge upwind | 6 ± 2 |
| --- | --- |
| Total steps upwind | 15 ± 2 |
| Total steps downwind | 1.3 ± 1.4 |
| Total steps to the right | 15 ± 3 |
| Total steps to the left | 18 ± 6 |

does indeed show degraded performance (see *Figure 5—figure supplement 1*). A systematic optimization of odor representation requires a considerable reformulation of the algorithm, which is beyond the scope of the current work. Note that, exclusively for the learned recovery, the optimal policy is enriched in actions downwind to avoid overshooting the source. Indeed, from positions beyond the source, the learned strategy is unable to recover the plume as it mostly casts sideways, with little to no downwind action. Intuitively, the precise locations where agents move downwind may be crucial to efficiently avoid overshooting. Thus, the policy may depend on specific details of the odor plume, consistent with poorer generalization of the learned recovery (discussed next). We expect that in conditions where overshooting the source is more prominent, downwind motion may emerge as an effective component of the recovery strategy, similar to observations in insects (e.g. *Wolf and Wehner, 2000*; *Álvarez-Salvado et al., 2018*).

The void state shows the most relevant differences: for both heuristic recoveries, 40% or more of the agents are in the void state and they are spatially spread out. In contrast, in the case of learned

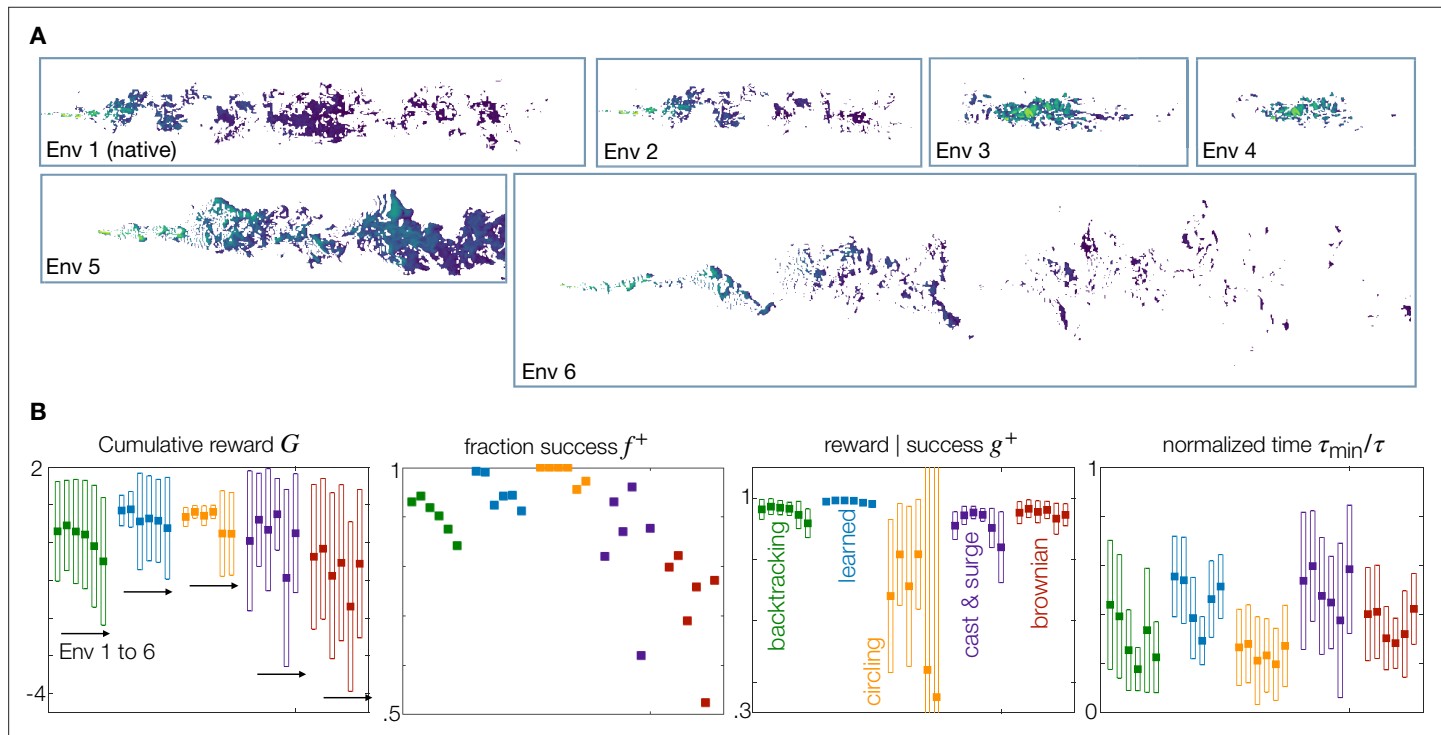

**Figure 5.** Generalization to statistically different environments. (**A**) Snapshots of odor concentration normalized with concentration at the source, color-coded from blue (0) to yellow (1) for environment 1–6 as labeled. Environment 1 is the native environment where all agents are trained. (**B**) Performance for the five recovery strategies backtracking (green), learned (blue), circling (orange), zigzag (purple) and brownian (red), with adaptive memory, trained on the native environment and tested across all environments 1–6. Four measures of performance defined in the main text are shown. Dark squares mark the mean, and empty rectangles ± standard deviation. For definition of the metrics used, see Materials and Methods, Agents Evaluation.

The online version of this article includes the following figure supplement(s) for figure 5:

**Figure supplement 1.** The learned recovery with adaptive memory and a single non-empty olfactory state (empty circles) displays degraded performance with respect to the full model (full circles).

recovery, the optimal policy limits the occurrence of the void state to 26% of the agents, confined to a narrow band near the edge of the plume. From these locations, the agents quickly recover the plume, explaining the boost in performance discussed above. In all distinct trainings of the agents with learned recovery, we observed that the trajectories in the void start with an initial surge of 6±2 steps; continue with either crosswind direction and then switch to the other side, with little to no downwind actions (see *Figure 4—figure supplement 2* and *Table 1*). This recovery thus mixes aspects of exploitation (surge) to aspects of exploration (cast): we defer a more in-depth analysis that disentangles these two aspects elsewhere.

## Tuning for adaptation to different environments

Finally, we test the performance of the trained agents on six environments, characterized by distinct fluid flows and odor plumes (*Figure 5* and Materials and methods). Environment 1 ('bulk native') is the environment where the agents were originally trained; Environment 2 ('bulk sparser') is obtained by increasing the threshold of detection, which makes the signals considerably more sparse with longer blanks. Environments 3 ('surface') and 4 ('surface sparser') are closer to the lower surface of the simulated domain, where the plume is smaller and fluctuates less. Environment 5 ('bulk lower Re') is a similar geometry, but obtained for a smaller Reynolds number and a different way to generate turbulence. Finally, Environment 6 ('bulk higher Re') has an even larger Reynolds number, a longer domain, and a smaller source, which creates an even more dramatically sparse signal. All bulk environments (1, 2, 5, and 6) are representative of conditions encountered far from a substrate, for example by flying or swimming organisms. The two surface environments (3 and 4) represent odor near surfaces relevant to terrestrial or benthic navigation (but not directly applicable to trail tracking, where odor traces *on* the substrate are tracked). Note that we consider a Schmidt number Sc = 1 appropriate for odors in air but not in water. However, we expect a weak dependence on the Schmidt number as the Batchelor and Kolmogorov scales are below the size of the source and we are interested in the large scale statistics (*Falkovich et al., 2001*; *Celani et al., 2014*; *Duplat et al., 2010*).

We consider agents with adaptive memory and compare the five recovery strategies discussed above – backtracking, learned, circling, cast and surge, and Brownian, see *Figure 5B*. Comparing performance across environments, we find that: (*i*) although performance is degraded when testing in non-native environments, backtracking, learned and circling recoveries with adaptive memory are still extremely likely to find the source. The upshot of generalization is that agents may navigate distinct turbulent plumes using a baseline strategy learned in a specific plume. Importantly, as most of these agents still do reach the source, fine-tuning may enable efficient adaptation to different environments. Further work is needed to establish how much fine-tuning is needed to fully adapt to different environments. (*ii*) Brownian and cast and surge recoveries have the lowest performance and generalization across all environments. Cast and surge is often used as a comparison (*Verano et al., 2023*) and can be extremely effective: our results do not contradict the literature, but simply showcase that cast width and surge length need to be carefully defined. (*iii*) The cumulative reward $G$ shows that the learned recovery is the best at generalizing to environments 2, 5, and 6. Particularly, in the most intermittent Environment 6 a striking 91% of agents succeed in finding the source, with trajectories less than twice as long as the shortest path to the source. (*iv*) Circling is the best at generalizing to environments 3 and 4, representative of less intermittent regions near the substrate. As observed for the native environment, circling favors success rate (metrics $f^+$) against speed (metrics $g^+$ and $\tau_{\min}/\tau$).

## Discussion

In this work, we showed that agents exposed to a turbulent plume learn to associate salient features of the odor time trace – the olfactory state – to an optimal move that guides them to the odor source. The upshot of responding solely to odor is that the agent does not navigate based on *where* it believes the target is and thus needs no map of space nor prior information about the odor plume, which avoids considerable computational burden. The only spatial awareness needed to implement this algorithm is the ability to orient motion relative to the mean flow, which is assumed known. In reality, animals cannot measure the mean flow but rely on local measures of flow speed, using for example antennas for insects (*Reynolds et al., 2010*; *Bell and Kramer, 1979*; *Suver et al., 2019*; *Okubo et al., 2020*), whiskers for rodents (*Yu et al., 2016*) or the lateral line for marine organisms

(*Liao, 2006*). Further work is needed to bridge the gap with our simplified setting. On the flip side, in our stimulus-response algorithm, agents need to start from within the plume, however sparse and fragmented. Indeed, far enough from the source, Q-learning agents are mostly in the void state and they can only recover the plume if they have previously detected the odor or are right outside the plume. In contrast, agents using a map of space can navigate from larger distances than are reachable by responding directly to odor cues. Indeed, in the map-based POMDP setting, absence of odor detection is still informative and it enables agents to first find the plume and then refine the search to localize the target within the plume (*Rigolli et al., 2022b*; *Loisy and Heinonen, 2023*).

We show that because the odor signal within a turbulent plume constantly switches on and off, navigation must handle both absence and presence of odor stimuli. We address this fundamental issue by alternating between two distinct strategies: (*i*) Prolonged absence of odor prompts entry in the void state and triggers a recovery strategy to make contact with the plume again. We explored four heuristic recoveries and found that spiraling around the location where the agent last detected odor is the most efficient heuristic, which privileges reliable success rather than speed. An even more efficient recovery can be learned that resembles cross-wind casting and limits the void state to a narrow region right outside of the plume. Casting is a well-studied computational strategy (*Baker, 1990*; *Balkovsky and Shraiman, 2002*) also observed in animal behavior, most famously in flying insects (*David et al., 1983*). Intriguingly, cast and surge also emerges in algorithms making use of a model of the odor, whether for Bayesian updates or for policy optimization (*Vergassola et al., 2007*; *Rigolli et al., 2022b*; *Verano et al., 2023*). Whether natural casting behavior is learned, as in Q-learning, or is hard-wired in a model of the odor plume remains a fascinating question for further research. Clearly, the width of the casts and length of the surges are of crucial importance: a hard-wired cast and surge recovery with arbitrary parameters shows poor performance. (*ii*) Odor detections prompt entry in non-void olfactory states, which predominantly elicit upwind surge. Blanks shorter than the sensing memory are ignored, that is agents do not enact recovery but respond to stimuli experienced prior to the short blank. The non-void olfactory states are crafted based on biologically plausible features which have been shown to harness positional information. Further work may optimize these non-void olfactory states by feature engineering, for example testing different discretizations to reduce redundancy. A drastic reduction to a single non-void olfactory state degrades performance, suggesting screening a large library of features using supervised learning as in *Rigolli et al., 2022a* may be used to potentially improve performance. This approach will provide a systematic ranking of the most efficient temporal features of odor time traces for navigation; however, it will have to test different memories, discretizations, and regression algorithms as well, making it cumbersome. Alternatively, feature engineering may be bypassed altogether by the use of recurrent neural networks (RNNs) (*Singh et al., 2023*) or finite state controllers (*Verano et al., 2023*). These algorithms are appealing in that they bypass entirely the need to hand-craft explicit features of odor time traces. On the flip side, they provide no explicit handle on the odor features that drive behavior nor on the specific duration of the temporal memory and how it is related to the physics of the odor cues. Thus, to extract these information, extra work is needed to interrogate these algorithms. For example, principal component analysis in *Singh et al., 2023* suggests the hidden state of trained agents correlates with biologically relevant variables, including head direction, odor concentration, and time since last detection. Finally, a systematic comparison using a common dataset is needed to elucidate how other heuristic and normative model-free algorithms handle odor presence *vs* odor absence.

To switch between the odor-driven strategy and the recovery strategy, we introduce a timescale $T$, which is an explicit form of temporal memory. $T$ delimits a sensing window extending in the recent past, prior to the present time. All odor stimuli experienced within the sensing window affect the current response. By using fixed memories of different durations, we demonstrate that an optimal memory exists and that the optimal memory minimizes the occurrence of the void state. On the one hand, long memories are detrimental because they delay recovery from accidentally exiting the plume. On the other hand, short memories are detrimental because they trigger recovery unnecessarily, i.e. even for blanks typically experienced within the turbulent plume. The optimal memory thus matches the typical duration of the blanks. To avoid using prior information on the statistics of the odor, we propose a simple heuristic setting memory adaptively equal to the most recent blank experienced along the path. The adaptive memory nears optimal performance despite dramatic fluctuations dictated by turbulence. Success of the heuristics suggests that a more accurate estimate of the future

blank time may enable an even better adaptive memory; further work is needed to corroborate this idea.

Thus, in Q-learning, memory is a temporal window matching odor blanks and distinguishing whether agents are in or out of the plume. The role of memory for olfactory search has been recently discussed in *Verano et al., 2023*. In POMDPs, memory is stored in a detailed belief of agent position relative to the source. In finite state controllers, memory denotes an internal state of the agent and was linked to a coarse-grained belief of the searcher being within or outside of the plume, similar to our findings. In recurrent neural networks, memory is stored in the learned weights. A quantitative relationship between these different forms of memory and their connection to spatial perception remains to be understood.

We conclude by listing a series of experiments to test these ideas in living systems. First, olfactory search in living systems displays memory (*Verano et al., 2023*; *Baker et al., 2018* and references therein). In insects, temporal scales can be measured associated with memory. Indeed, for flying insects, loss of contact with a pheromone plume triggers crosswind casting and sometimes even downwind displacement (*Cardé, 2021*; *Kuenen and Cardé, 1994*). Interestingly, the onset of casting is delayed with respect to loss of contact with the plume (*Kuenen and Cardé, 1994*; *van Breugel and Dickinson, 2014*), but this delay is not understood. Similarly, an odor detection elicits upwind surges that can outlast the odor by several seconds (*Kathman et al., 2024*; *Álvarez-Salvado et al., 2018*). In walking flies, the timing of previous odor encounters biases navigation (*Demir et al., 2020*). (How) do these temporal timescales depend on the waiting times between previous detections? Using optogenetics (*Gepner et al., 2015*; *Hernandez-Nunez et al., 2015*; *Matheson et al., 2022*; *Stupski and van Breugel, 2024b*) or olfactory virtual reality with controlled odor delivery (*Radvansky and Dombeck, 2018*), experiments may measure memory as a function of the full history of odor traces. For insects, one may monitor memory by tracking the onset of crosswind casting with respect to the loss of the plume. More in general, a temporal memory may be defined by monitoring how far back in the past two odor traces should be identical in order to elicit the same repertoire of motor controls.

Second, our algorithm learns a stimulus-response strategy that relies solely on odor cues. The price to pay is that the agent must follow the ups and downs of the odor trace in order to compute averages and recognize blanks. A systematic study may use our algorithm to test the requirements of fidelity of this temporal representation and how it depends on turbulence. How does turbulence affect the fidelity of odor temporal representation in living systems? Crustaceans provide an excellent model system to ask this question, as they are known to use bursting olfactory receptor neurons to encode temporal information from olfactory scenes (*Bobkov and Ache, 2007*; *Ache et al., 2016*). Temporal information is also encoded in the olfactory bulb of mammals (*Carey et al., 2009*; *Ackels et al., 2021*). Organisms with chemo-tactile systems like the octopus (*Allard et al., 2023*) may serve as a comparative model, to ask whether touch-chemosensation displays a sloppier temporal response, reflecting that surface-bound stimuli are not intermittent.

Third, our Q-learning algorithm requires the agent to receive olfactory information, thus start near or within the odor plume. In contrast, algorithms making use of a spatial map and prior information on the odor plume may first search for the plume (in conditions of near zero information) and then search the target within the plume (*Rigolli et al., 2022b*; *Loisy and Heinonen, 2023*; *Vergassola et al., 2007*). Animals are known to use prior information to home into regions of space where the target is more likely to be found; but they can switch to navigation in response to odor (see e.g. *Cardé, 2021*; *Schal, 1982*; *Gire et al., 2016*; *Baker et al., 2018*). What triggers the switch from spatial navigation driven by prior information to sensory driven navigation using odor? For mice, the need for spatial perception may be tested indirectly by comparing paths in light *vs* dark, noting that neuronal place fields, that mediate spatial perception, are better stabilized by vision than olfaction (*Save et al., 2000*; *Zhang and Manahan-Vaughan, 2015*). Thus in the light, animals have the ability to implement both map-less and map-based algorithms, whereas in the dark they are expected to more heavily rely on map-less algorithms. To make sure animals start searching for the odor target even before sensing odor, operant conditioning can be deployed so that animals associate an external cue (e.g. a sound) to the beginning of the task. Note that distinct species control locomotion differently, and as a result, trajectories are usually far more complex than a sequence of discrete steps on a checkerboard. Thus, to compare algorithms to animal behavior, a more detailed model of the specific motor controls is to be developed.

The reinforcement learning view of olfactory navigation offers an exciting opportunity to probe how living systems interact with the environment to accomplish complex real-world tasks affected by uncertainty. Coupling time-varying odor stimuli with spatial perception is an instance of the broader question asking how animals combine prior knowledge regarding the environment with reaction to sensory stimuli. We hope that our work will spark further progress into connecting these broader questions to the physics of fluids.

# Materials and methods

## Key resources table

| Reagent type (species) or resource | Designation | Source or reference | Identifiers | Additional information |
|---|---|---|---|---|
| Software, algorithm | Computational fluid dynamics source code | *Viola et al., 2023*; *Viola et al., 2020*; *Viola et al., 2022*; *Verzicco et al., 2025*. Courtesy of F. Viola. | https://gitlab.com/vdv9265847/IBbookVdV/ | Reused Computational Fluid Dynamics software used to run simulations of odor transport. An earlier version of the code is publicly available at the website indicated in the 'identifiers' entry, described in *Verzicco et al., 2025*. Our simulations were conducted using a GPU-accelerated version of the code that was developed by F. Viola and colleagues in the Refs indicated in the 'Source or reference' entry. This version will be shared in the near future. All requests may be directed to Viola and colleagues |
| Software, algorithm | Datasets of odor field obtained with computational fluid dynamics – Environments 5 and 6 | This paper | https://doi.org/10.5281/zenodo.14655991 | Newly developed datasets of turbulent odor fields, obtained through computational fluid dynamics. |
| Software, algorithm | Tabular Q-learning | This paper, *Rando, 2025* | https://github.com/Akatsuki96/qlearning_for_navigation | Newly developed Model-free Algorithm for training olfactory search agents, with settings described in materials and methods. Shared on github address mentioned as 'Identifier' |
| Software, algorithm | Datasets of odor field obtained with computational fluid dynamics – Environments 1–4 | *Rigolli et al., 2022b* | https://doi.org/10.5281/zenodo.6538177 | Reused datasets of turbulent odor fields, obtained through computational fluid dynamics in *Rigolli et al., 2022b* |

## Data description

The data we used to train the agents (Environment 1) is a set of 2598 matrices $\{D_t\}_{t=1}^{2598}$. Every matrix $D_t \in \mathbb{R}^{1225 \times 280}$ contains the odor intensity in every position $(i, j)$ i.e. $(D_t)_{i,j}$ represents the odor intensity in position $(i, j)$ at time $t$. The source of odor is in position $(20, 142)$, and in order to simplify the

**Table 2.** Gridworld geometry.
From Top: 2D size of the simulation, agents that leave the simulation box continue to receive negative reward and no odor; number of time stamps in the simulation, beyond which simulations are looped; number of actions per time stamp; speed of the agent; noise level below which odor is not detected; location of the source on the grid. See *Table 3* for the values of the grid size $\Delta x$ and time stamps at which odor snapshots are saved.

| | Simulation 1 | Simulation 2 | Simulation 3 |
|---|---|---|---|
| 2D simulation grid | 1225 × 280 | 1024 × 256 | 2000 × 500 |
| # time stamps | 2598 | 5000 | 5000 |
| # decisions per time stamp | 1 | 1 | 1 |
| Speed (grid points / time stamp) | 10 | 10 | 10 |
| $n_{\text{lvl}}$ | $3 \times 10^{-6}$ | $3 \times 10^{-6}$ | $10^{-4}$ |
| Source location | (20, 142) | (128, 128) | (150, 250) |

**Table 3.** Parameters of the simulations.

From Left to Right: Simulation ID (1, 2, 3); Length $L$, width $W$, height $H$ of the computational domain; mean horizontal speed $U_b = \langle u \rangle$; Kolmogorov length scale $\eta = (\nu^3/\epsilon)^{1/4}$ where $\nu$ is the kinematic viscosity and $\epsilon$ is the energy dissipation rate; mean size of gridcell $\Delta x$; Kolmogorov timescale $\tau_\eta = \eta^2/\nu$; energy dissipation rate $\epsilon = \nu/2\langle(\partial u_i/\partial x_j + \partial u_j/\partial x_i)^2\rangle$; wall unit $y^+ = \nu/u_\tau$ where $u_\tau$ is the friction velocity; bulk Reynolds number $Re_b = U_b(H/2)/\nu$ based on the bulk speed $U_b$ and half height; magnitude of velocity fluctuations $u'$ relative to the bulk speed; large eddy turnover time $T = H/2u'$ frequency at which odor snapshots are saved $\times \tau_\eta$. For each simulation, the first row reports results in non-dimensional units. Second and third rows provide an idea of how non-dimensional parameters match dimensional parameters in real flows in air and water, assuming the Kolmogorov length is 1.5 mm in air and 0.4 mm in water.

| Sim ID | $L$ | $W$ | $H$ | $U_b$ | $\eta$ | $\Delta x$ | $\tau_\eta$ | $\epsilon$ | $y^+$ | $Re_b$ | $\frac{u'}{U_b}$ | $T$ | $\omega_{save} \times \tau_\eta$ |
|---|---|---|---|---|---|---|---|---|---|---|---|---|---|
| 1 | 40 | 8 | 4 | 23 | 0.006 | 0.025 | 0.01 | 39 | 0.0035 | 11500 | 15% | $64\tau_\eta$ | 1 |
| air | 9.50 m | 1.90 m | 0.96 m | $36\frac{\text{cm}}{\text{s}}$ | 0.15 cm | 0.6 cm | 0.15 s | $6.3e-4\frac{\text{m}^2}{\text{s}^3}$ | 0.09 cm | | | | |
| water | 2.66 m | 0.53 m | 0.27 m | $8.6\frac{\text{cm}}{\text{s}}$ | 0.04 cm | 0.2 cm | 0.18 s | $3e-5\frac{\text{m}^2}{\text{s}^3}$ | 0.02 cm | | | | |
| 2 | 20 | 5 | 2 | 14 | 0.004 | 0.02 | 0.005 | 163 | 0.0038 | 7830 | 15% | $95\tau_\eta$ | 5 |
| air | 7.50 m | 1.875 m | 0.75 m | $17.5\frac{\text{cm}}{\text{s}}$ | 0.15 cm | 0.75 cm | 0.15 s | $6.3e-4\frac{\text{m}^2}{\text{s}^3}$ | 0.142 cm | | | | |
| water | 2.00 m | 0.50 m | 0.20 m | $3.9\frac{\text{cm}}{\text{s}}$ | 0.04 cm | 0.2 cm | 0.18 s | $3e-5\frac{\text{m}^2}{\text{s}^3}$ | 0.038 cm | | | | |
| 3 | 20 | 5 | 2 | 22 | 0.0018 | 0.01 | 0.0025 | 204 | 0.0012 | 17500 | 13% | $141\tau_\eta$ | 2.5 |
| air | 16.7 m | 4.18 m | 1.67 m | $30.6\frac{\text{cm}}{\text{s}}$ | 0.15 cm | 0.83 cm | 0.15 s | $6.3e-4\frac{\text{m}^2}{\text{s}^3}$ | 0.1 cm | | | | |
| water | 4.44 m | 1.11 m | 0.44 m | $6.8\frac{\text{cm}}{\text{s}}$ | 0.04 cm | 0.22 cm | 0.18 s | $3e-5\frac{\text{m}^2}{\text{s}^3}$ | 0.03 cm | | | | |

training, we considered as terminal states every position in a circle centered in the source position and with radius 10 called the *source region*. Data are obtained from a direct numerical simulation of the Navier-Stokes equations and the equations of transport of the odor. Environments 1–4 are derived from Simulation 1, a direct numerical simulation of a channel flow described in *Rigolli et al., 2022a* and used to develop a POMDP algorithm in *Rigolli et al., 2022b*, dataset available from *Rigolli et al., 2022c*. The simulation represents a boundary layer whose dimensions match the lowest ~1 m of the atmosphere and with horizontal dimensions ~1.9 × 9.5 m and Reynolds number $Re_\lambda \sim 1400$ (on the low side of atmospheric Reynolds typically ranging from 1000–10,000 *Gulitski et al., 2007*). Odor snapshots are extracted at a height $\sim .5$ m from the ground for Environments 1 and 2 and $\sim .01$ m for Environments 3 and 4 respectively. We preprocess the data to zero every entry of these matrices when they are smaller than a *noise level* $n_{lvl} := 3 \times 10^{-6}$ (or 0.13% relative to concentration at the source). The noise level is increased to 0.22% in Environments 2 and 4. Data information are summarized in *Tables 2 and 3*. Levels of intermittency in *Figure 1* show that only a thin core region has intermittency larger than 66%, whereas the most challenging regions at the edge of the plume have intermittency under 33%. For reference, experimental values of 25% to 20% were reported for a surrogate odor in the atmospheric boundary layer, along the centerline at 2–15 m from the source (*Murlis and Jones, 1981*).

Environments 5 and 6 correspond to horizontal slices at mid height extracted from two additional simulations we performed to corroborate the results (Simulation 2 and 3). In Simulation 2, the odor is advected by a turbulent open channel flow, with three hemispherical obstacles placed on the ground close to the inlet to generate turbulence. The Navier-Stokes *equations (1)* and advection-diffusion equation for odor transport (3) are solved using a central second-order finite difference scheme. The convective terms are discretized in time using an explicit Adams–Bashforth method, and the viscous and diffusion terms using an implicit Crank-Nicolson method (*Viola et al., 2023*; *Viola et al., 2020*; *Viola et al., 2022*). The code is written in Fortran and is GPU parallelized. The channel is divided into 1024 × 256 × 128 grid points along streamwise, spanwise, and wall-normal directions respectively.

The corresponding average spatial resolutions are $\Delta x = 5\eta$, $\Delta y = 5\eta$, $\Delta z = 4\eta$, where $\eta$ is the Kolmogorov length scale. Three hemispheres of radius $100\eta$ are placed at a distance of $250\eta$ from the inlet on the ground, equally spaced along the spanwise direction. The obstacles are implemented using the immersed boundary method (**Verzicco et al., 2025**). The channel is forced using a constant pressure gradient. For the velocity field, we impose a no-slip boundary condition at the ground and on the obstacles ($\boldsymbol{u} = 0$) and a free-slip boundary on top ($u_z = 0$, $\partial_z u_x = \partial_z u_y = 0$). The velocity field is periodic along the streamwise and spanwise directions. The bulk Reynolds number is 7800. For the odor field, we impose Dirichlet condition ($c = 0$) at the ground, on the obstacles and inlet, no-flux ($\partial_z c = 0$) on top, and outflow along other directions. Similar to the native environment, we choose the Schmidt number to be 1. The odor source is located downstream of the obstacle and centered at [$640\eta$, $640\eta$, $256\eta$] along streamwise, spanwise, and wall-normal directions. respectively. The odor source has a Gaussian profile with a standard deviation of $8\eta$.

Simulation 3 is similar to Simulation 2, albeit with a higher bulk Reynolds number of 17,500. Here, the channel is divided into 2000 × 500 × 200 grid points and has an average spatial resolution of $\Delta x = \Delta y = \Delta z = 5.5\eta$. The odor source has a Gaussian profile centered at [$825\eta$, $1375\eta$, $550\eta$] with a standard deviation of $3\eta$. For Environments 5 and 6, the noise level is 0.01% relative to concentration at the source. See Table for a summary of parameters and how they match the physical dimensions of the domain.

$$\rho \left( \frac{\partial u}{\partial t} + \boldsymbol{u}.\boldsymbol{\nabla u} \right) = -\nabla P + \mu \boldsymbol{\nabla}^2 u + \boldsymbol{f};$$

$$\boldsymbol{\nabla u} = 0. \tag{1}$$

$$\frac{\partial z}{\partial t} + \mathbf{u}.\nabla z = D\nabla^2 z + s. \tag{2}$$

## Olfactory states, features, and discretization

Each agent stores the odor concentrations detected in the previous $T$ time steps in a vector $\mathbf{M} = (z(s(t - T), t - T), ..., z(s(t), t))$. We introduce an adaptive sensitivity threshold function $s_{\text{thr}}(\cdot)$ defined as

$$s_{\text{thr}}(T) := \max \left\{ \frac{C_{\text{thr}}}{T} \sum_{i=1}^{T} M_i, n_{\text{thr}} \right\}, \tag{3}$$

where $M_i$ denotes the $i$-th element of $M$ and $C_{\text{thr}} > 0$ is a scaling constant (in our experiments we set it as 0.5). $T$ denotes the cardinality of $M$. Given a memory $M$, we can define the filtered memory $\Delta^M$ as the set which contains every element of the memory $M$ that is higher than the sensitivity threshold $s_{\text{thr}}(M)$, that is

$$\Delta^M := \{z \in M \mid z > s_{\text{thr}}(M)\}. \tag{4}$$

Then at time step $t$, given the agent memory $M_t$, we define the average intensity $c(M_t)$ and the intermittency $i(M_t)$ as:

$$c(M_t) := \begin{cases} \frac{1}{|\Delta^{M_t}|} \sum_{i=1}^{|\Delta^{M_t}|} \left( \Delta^{M_t} \right)_i, & |\Delta^{M_t}| > 0 \\ \\ 0 \end{cases}, \tag{5}$$

$$i(M_t) := \frac{|\Delta^{M_t}|}{|M_t|}.$$

Note that the average intensity is defined on the filtered memory $\Delta^M$, that is conditioned to detecting odors above threshold. Since the features defined in **Equation 6** return real numbers, in order to use (tabular) q-learning, we need to discretize them. We denote with $\bar{i}(M_t)$ the discretized intermittency. This is defined as follow

$$\bar{i}(M_t) := \begin{cases} 0, & \text{if } i(M_t) \leq 0.33 \\ 1, & \text{if } 0.33 < i(M_t) \leq 0.66 \\ 2, & \text{if } i(M_t) > 0.66 \end{cases} \tag{6}$$

The average intensity is bounded between zero and the maximum concentration of odor at the source. To avoid prior information on the source, we use a more structured procedure to discretize the average intensity online, based on the agent's experience only. At every time step $t$, the average intensity $c(M_t)$ is computed and collected in a dataset $X_t$, that is

$$X_t := \{c(M_0), \cdots, c(M_t)\}.$$

Then, its discretized value is obtained by the following rule:

$$\bar{c}(M_t, X_t) := \begin{cases} 0, & c(M_t) \leq p(X_t, 25) \\ 1, & p(X_t, 25) < c(M_t) \leq p(X_t, 50) \\ 2, & p(X_t, 50) < c(M_t) \leq p(X_t, 80) \quad, \\ 3, & p(X_t, 80) < c(M_t) \leq p(X_t, 99) \\ 4, & c(M_t) > p(X_t, 99) \end{cases} \tag{7}$$

where $p(X_t, n)$ denotes the $n$-th percentile of $X_t$. Finally, we can define the feature map $\phi_t$ as a function of the memory $M_t$ and the dataset of average intensities $X_t$ at time step $t$

$$\phi_t(M_t, X_t) := [\bar{i}(M_t), \bar{c}(M_t, X_t)].$$

This defines the current olfactory state $s_t$ that is at time step $t$, the agent is in the olfactory state $o_t := \phi_t(M_t, X_t)$. The case where the agent has no odor detections above threshold in its current memory, that is $|\Delta(M_t)| = 0$ corresponds to an additional state called void state ($\emptyset$) in the main text.

## Agent behavior and policies

Now, we describe how the agent interacts with the environment to solve the navigation problem. At every time step $t \in \mathbb{N}$, the agent observes an odor point $z_t$ and updates its memory, including the new observation and removing the oldest that is it defines a memory $M_t$ with the following rule

$$M_t := \left[ (M_{t-1})_2, \cdots, (M_{t-1})_{|M_{t-1}|}, o_t \right]. \tag{8}$$

Then, it updates the dataset of average intensities that is $X_t := X_{t-1} \cup \{c(M_t)\}$ and it computes the olfactory state $o_t$. According to $o_t$, the agent chooses an action $a_t$ using a policy. As indicated in the main text, actions are the coordinate directions that is we define an action set $\mathcal{A}$ as follow

$$\mathcal{A} := \{e_1, e_2, -e_1, -e_2\},$$

where $e_i$ denotes the $i$-th canonical base. Actions are steps in any of the four directions, labeled relative to the mean flow which is assumed fixed and known. The gridworld is infinite, in that agents can leave indefinitely. If they exit the simulation box, they continue to receive zero signal and negative reward –0.001. As explained in the main text, actions are selected using one of two policies according to the current olfactory state $o_t$. More precisely, if the olfactory state $o_t$ is not the void state, then the ($\epsilon$-greedy) Q-learning policy is used. Formally, let $Q$ be the Q matrix of the agent and let $o_t \neq \emptyset$, then the agent plays the action $a_t$ such that

$$a_t = \begin{cases} a \in \arg\max_{a \in \mathcal{A}} Q(o_t, a) & \text{with probability } 1 - \epsilon \\ a \sim \mathcal{U}(\mathcal{A}) & \text{with probability } \epsilon \end{cases}, \tag{9}$$

where, with $a \sim \mathcal{U}(\mathcal{A})$, we indicate an action $a$ uniformly sampled from $\mathcal{A}$. At the test phase, the exploration-exploitation parameter $\epsilon$ is set to 0, and thus, in an olfactory state $o_t \neq \emptyset$ the policy is deterministic. While training phase behavior is described in the next paragraphs. In the void state $o_t = \emptyset$, the agent chooses the action $a_t \in \mathcal{A}$ according to a separated policy called *recovery strategy*.

In our experiments, we defined and compared three different recovery strategies: Brownian, Backtracking, and Learned.

### Brownian recovery

It is the simplest strategy we consider, consisting of playing random actions in the void state. Suppose that at time step $t$, the agent is in the void olfactory state, that is $o_t = \emptyset$, then $a_t$ is sampled uniformly from the action set $\mathcal{A}$. However, it is important to note that long-memory agents start to recover when they are already far from the plume, and hitting the plume by random walk is prohibitively long. To avoid wandering away from the plume, the memory is constrained to be shorter, consistent with the observation that the optimal memory is $T^* = 3$ to $5$, much shorter than for backtracking. At this memory, several blanks within the plume will cause the agent to recover, hence the lower performance of the Brownian recovery.

### Backtracking recovery

In order to accelerate recovery from accidentally exiting the plume, we let the agents backtrack to the position where they last detected the odor. To this end, we first enumerate the actions with numbers from one to four. Then, we introduce a new memory called *action memory* $A$. For simplicity, we consider the setting in which $|A| = |M|$. At time step $t = 0$, this memory is initialized as a vector of zeros indicating that the action memory is empty that is we define $A_0 \in \mathbb{N}^{|M|}$ such that for every $i = 1, \cdots, |A|$

$$A_i = 0.$$

For every time step $t > 0$, the agent observes an odor point $z_t$ and updates the memory through (*Equation 8*). Moreover, the action memory is updated according to the status of the memory. If the last observation is smaller than the sensitivity threshold, that is $z_t < s_{\text{thr}}(M_t)$, the action previously played $a_{t-1}$ (represented by a natural number in $[1, 4]$) is stored in the action memory, that is for some $\Delta > 0$, let

$$A_{t-1} = [a_{t-\Delta}, \cdots, a_{t-2}, 0, \cdots, 0].$$

Then

$$A_t = [a_{t-\Delta}, \cdots, a_{t-2}, a_{t-1}, \cdots, 0].$$

If at time step $t$, the observation $z_t$ is larger than the sensitivity threshold then the action memory is reset, that is $A_t \in \mathbb{N}^{|M|}$ with $(A_t)_i = 0$ for every $i$. If at time step $t$, the memory is empty, that is $c(M_t) = 0$, then the backtracking procedure is executed: the last non-zero element of the action memory is extracted, and the inverse action is played that is for some $\Delta > 0$, let

$$A_{t-1} = [a_{t-\Delta}, \cdots, a_{t-2}].$$

Then, it plays the action $a_{t-2}$ and updates the action memory as follow

$$A_t = [a_{t-\Delta}, \cdots, a_{t-3}, 0].$$

This procedure is repeated until either an observation larger than the sensitivity threshold is obtained or the action memory becomes empty. In the former case, the action memory is cleared, and the action is chosen according to the Q-learning policy (*Equation 9*). In the latter case, a random action is played.

Note that this strategy only provides exploration after the backtracking fails to recover detections. Also, if agents start with no detection at time 0, the procedure is equivalent to Brownian motion.

### Circling recovery

In this case, the recovery strategy consists of adopting a circling behavior (*Stupski and van Breugel, 2024a*) when the agent is in a void state. The agent keeps in memory two counters, $t_{\text{void}}$ and $T_{\text{change}}$, as well as an action $a_{\text{void}} \in \mathcal{A}$, initialized to 0, 1, and $e_1$, respectively. The first counter represents the consecutive number of void observations, $a_{\text{void}}$ is the action to play when the void state is reached, and $T_{\text{change}}$ is a time threshold that indicates when to switch the action $a_{\text{void}}$. When the agent reaches

a void state, it plays the action $a_{\text{void}}$ and increments the counter $t_{\text{void}}$ by one. If $t_{\text{void}} = T_{\text{change}}$ then the agent resets $t_{\text{void}}$ to zero, increases $T_{\text{change}}$ by one, and updates the action $a_{\text{void}}$ as follows

$$a_{\text{void}} \leftarrow \begin{cases} e_2 & \text{if } a_{\text{void}} = e_1 \\ -e_1 & \text{if } a_{\text{void}} = e_2 \\ -e_2 & \text{if } a_{\text{void}} = -e_1 \\ e_1 & \text{if } a_{\text{void}} = -e_2 \end{cases}$$

When the agent receives a non-void observation, it resets $t_{\text{void}}$ to 0, $T_{\text{change}}$ to 1, and $a_{\text{void}}$ to $e_1$.

### Cast-surge recovery

In this case, the agent plays a cast-surge behavior (**Baker, 1990**) when reaching the void state. As in the circling recovery strategy, the agent keeps two counters, $t_{\text{void}}$ and $T_{\text{change}}$, as well as an action $a_{\text{void}} \in \mathcal{A}$, initialized to 0, 1, and $e_2$, respectively. At every step in the void state, the agent plays the action $a_{\text{void}}$ and increments the void counter $t_{\text{void}}$ by one. If $t_{\text{void}} = T_{\text{change}}$, the agent takes an upwind step, doubles the value of $T_{\text{change}}$, updates $a_{\text{void}}$ by setting it to $-a_{\text{void}}$, and resets $t_{\text{void}}$ to zero. When the agent receives a non-void observation, it resets $t_{\text{void}}$ to 0, $T_{\text{change}}$ to 1, and $a_{\text{void}}$ to $e_2$.

### Learned recovery

In this case, the recovery policy is learned by splitting the void state into several states labeled by the time since entry in the void state. In our experiments, we split the void state into 50 states. Actions are then learned as in all other non-void states, and the optimal action is always chosen with (**Equation 9**).

### Training

An agent starts at a random location within the odor plume at time 0. Its memory is initialized with the prior $|M_0|$ odor detections at its initial location $M_0 = [z_{-|M_0|}, \cdots , z_0]$, obtained from the fluid dynamics simulation. The Q-function $Q_0$ is initialized with 0.6 for all actions and olfactory states. The first dataset of average intensities contains the first value $X_0 = \{c(M_0)\}$. At every time step $t > 0$, the agent gets an odor observation $z_t$ from its new position and updates its memory, including the new observation and removing the oldest; the olfactory state $o_t$ is computed (as described in previous paragraphs). The dataset of average intensities is updated: $X_t = X_{t-1} \cup \{c(M_t)\}$. Exploration-exploitation parameter $\epsilon_k$ is scheduled as follow

$$\epsilon_k = \eta_{\text{init}} \exp(-\eta_{\text{decay}} k),$$

where, in our experiments, $\eta_{\text{init}} = 0.99$ and $\eta_{\text{decay}} = 0.0001$. At every episode $k$, the Q-function is updated at every time step $t$ as

$$Q_{k+1}(s_t, a_t) := (1 - \alpha_k) Q_k(s_t, a_t) + \alpha_k(r_t + \gamma \max_{a'} Q_k(s_{t+1}, a')),$$

where $R_t$ is the immediate reward received playing the action $a_t$ and $o_{t+1}$ are the current and the next olfactory states and $\alpha_k$ is the learning rate at episode $k$. This is scheduled as

$$\alpha_k = \alpha_{\text{init}} \exp(-\alpha_{\text{decay}} k),$$

where, in our experiments, $\alpha_{\text{init}} = 0.25$ and $\alpha_{\text{decay}} = 0.001$. For the experiments, agents are trained in 100,000 episodes and a horizon of 5000 steps. The agent velocity is set to 10, and the discount factor is $\gamma = 0.9999$.

### Agents Evaluation

To evaluate the performance of the different agents, we consider four metrics: the cumulative reward $G$ (which is the actual quantity that the algorithm optimizes for); normalized time (defined below); the fraction of success $f^+$ and the value conditioned on success $g^+$. For a fixed position $(i, j)$, we denote with $\tau_{\text{min}}(i, j)$ the minimum number of steps required to reach the source region from $(i, j)$ that is the length of the shortest path.

We define $D_{\text{init}}$ the set of points in which the first observation is above the sensitivity threshold (valid points). For each initial position $(i,j) \in D_{\text{init}}$, let $\tau(i,j)$ be the duration of the path obtained by an agent to reach the source. Note that $\tau(i,j)$ is a random variable for the stochastic backtracking and Brownian recoveries, but it is deterministic for the learned strategy that has no random components. For each admissible location $(i,j)$, we define four performance metrics:

$$G(i,j) = \quad \langle e^{-\lambda \tau(i,j)} - \frac{\sigma}{1-\gamma}(1 - e^{-\lambda \tau(i,j)}) \rangle$$

$$f^+(i,j) = \frac{n_{\text{success}}(i,j)}{n_{\text{reps}}}$$

$$g^+(i,j) = \langle e^{-\lambda \tau(i,j)} \,|\, \text{success} \rangle$$

$$\frac{\tau_{\min}}{\tau}(i,j) = \langle \frac{\tau_{\min}(i,j)}{\tau(i,j)} \rangle$$

where $n_{\text{reps}}$ is the number of test trajectories from each admissible location, and we use $n_{\text{reps}} = 10$. We then compute statistics of the performance metrics over the $D_{\text{init}}$ initial positions and report the average ($\langle \cdot \rangle$) and standard deviation (std). Note that both the backtracking and Brownian strategies have stochastic steps; for these strategies, $f^+(i,j)$ denotes the average success fraction computed at each position over 10 repetitions.

## Acknowledgements

This research was supported by grants to AS from the European Research Council (ERC) under the European Union's Horizon 2020 research and innovation programme (grant agreement No 101002724 RIDING), the Air Force Office of Scientific Research under award number FA8655-20-1-7028, and the National Institutes of Health (NIH) under award number R01DC018789. LR and MR acknowledge the financial support of the European Research Council (grant SLING 819789), the European Commission (Horizon Europe grant ELIAS 101120237), the US Air Force Office of Scientific Research (FA8655-22-1-7034), the Ministry of Education, University and Research (FARE grant ML4IP R205T7J2KP; grant BAC FAIR PE00000013 funded by the EU - NGEU) and the Center for Brains, Minds and Machines (CBMM), funded by NSF STC award CCF-1231216. MR is a member of the Gruppo Nazionale per l'Analisi Matematica, la Probabilit'a e le loro Applicazioni (GNAMPA) of the Istituto Nazionale di Alta Matematica (INdAM). This work represents only the view of the authors. The European Commission and the other organizations are not responsible for any use that may be made of the information it contains. We thank Francesco Viola for sharing a GPU accelerated version of the CFD code as well as support and discussions regarding computational fluid dynamics; Antonio Celani, Venkatesh Murthy, Yujia Qi, Francesco Boccardo, Luca Gagliardi, Francesco Marcolli and Arnaud Ruymaekers for comments on the manuscript.

## Additional information

### Competing interests

Agnese Seminara: Reviewing editor, eLife. The other authors declare that no competing interests exist.

### Funding

| Funder | Grant reference number | Author |
| --- | --- | --- |
| European Research Council | 10.3030/101002724 | Martin James<br>Agnese Seminara |
| European Research Council | 10.3030/819789 | Marco Rando<br>Lorenzo Rosasco |
| National Institutes of Health | R01DC018789 | Martin James<br>Alessandro Verri<br>Agnese Seminara |

| Funder | Grant reference number | Author |
| --- | --- | --- |
| Air Force Office of Scientific Research | FA8655-20-1-7028 | Marco Rando<br>Martin James<br>Alessandro Verri<br>Lorenzo Rosasco<br>Agnese Seminara |
| Air Force Office of Scientific Research | FA8655-22-1-7034 | Marco Rando<br>Lorenzo Rosasco |
| Ministero dell'Istruzione, dell'Università e della Ricerca | ML4IP R205T7J2KP | Marco Rando<br>Lorenzo Rosasco |
| National Science Foundation | CCF-1231216, Center for Brain Minds and Machines | Lorenzo Rosasco |
| HORIZON EUROPE Framework Programme | 101120237 ELIAS | Marco Rando<br>Lorenzo Rosasco |
| NextGenerationEU | BAC FAIR PE00000013 | Marco Rando<br>Alessandro Verri<br>Lorenzo Rosasco |

The funders had no role in study design, data collection and interpretation, or the decision to submit the work for publication.

## Author contributions

Marco Rando, Conceptualization, Software, Investigation, Visualization, Methodology, Writing – original draft, Writing – review and editing; Martin James, Investigation, Visualization, Writing – original draft; Alessandro Verri, Conceptualization, Supervision, Funding acquisition, Writing – review and editing; Lorenzo Rosasco, Conceptualization, Supervision, Funding acquisition, Methodology, Writing – original draft, Project administration, Writing – review and editing; Agnese Seminara, Conceptualization, Supervision, Funding acquisition, Visualization, Methodology, Writing – original draft, Project administration, Writing – review and editing

## Author ORCIDs

Marco Rando ⓘ https://orcid.org/0009-0008-3839-1429
Agnese Seminara ⓘ https://orcid.org/0000-0001-5633-8180

Reviewer #1 (Public review): https://doi.org/10.7554/eLife.102906.3.sa1
Reviewer #2 (Public review): https://doi.org/10.7554/eLife.102906.3.sa2
Author response https://doi.org/10.7554/eLife.102906.3.sa3

# Additional files

## Supplementary files

MDAR checklist

## Data availability

The datasets and code used to perform the experiments are available at the following links: (1) Newly created source code for Q-learning training and test: https://github.com/Akatsuki96/qlearning_for_navigation (copy archived at *Rando, 2025*) (2) Newly created datasets of odor snapshots: https://doi.org/10.5281/zenodo.14655992. Additionally, the work re-uses previously developed data and codes: Reused datasets of odor snapshots from *Rigolli et al., 2022c*. Reused source code to run full computational fluid dynamics simulations: the code is described in *Verzicco et al., 2025* and an earlier version of the code is available at https://gitlab.com/vdv9265847/IBbookVdV/ (*Verzicco et al., 2024*). The GPU-accelerated version of the code was used here; it was developed in *Viola et al., 2023*; *Viola et al., 2020*; *Viola et al., 2022* and *Verzicco et al., 2025*, and was obtained as a courtesy of F. Viola. The full source code will be made available by the authors of the cited work in the near future.This is a computational study: no experimental data have been generated for this manuscript.

The following dataset was generated:

| Author(s) | Year | Dataset title | Dataset URL | Database and Identifier |
|---|---|---|---|---|
| Marco R, Martin J, Alessandro V, Lorenzo R, Agnese S | 2025 | Q-learning with temporal memory to navigate turbulence - Datasets | https://doi.org/10.5281/zenodo.14655992 | Zenodo, 10.5281/zenodo.14655992 |

The following previously published dataset was used:

| Author(s) | Year | Dataset title | Dataset URL | Database and Identifier |
|---|---|---|---|---|
| Rigolli N, Reddy G, Seminara A, Vergassola M | 2022 | Alternation emerges as a multi-modal strategy for turbulent odor navigation - Dataset | https://doi.org/10.5281/zenodo.6538177 | Zenodo, 10.5281/zenodo.6538177 |

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
