## [Editor Report · eLife Assessment]

This **important** study uses reinforcement learning to study how turbulent odor stimuli should be processed to yield successful navigation. The authors find that there is an optimal memory length over which an agent should ignore blanks in the odor to discriminate whether the agent is still inside the plume or outside of it, complementing recent studies using recurrent neural networks and finite state controllers to identify optimal strategies for navigating a turbulent plume. The strength of evidence is **compelling**, presenting a novel approach to understanding optimal representations for navigation in stochastic sensory environments.

---

## [Referee Report · Reviewer #1 (Public review)]

Overall I found the approach taken by the authors to be clear and convincing. It is striking that the conclusions are similar to those obtained in a recent study using a different computational approach (finite state controllers), and lends confidence to the conclusions about the existence of an optimal memory duration. There are a few questions that could be expanded on in future studies:

(1) Spatial encoding requirements

The manuscript contrasts the approach taken here (reinforcement learning in a gridworld) with strategies that involve a "spatial map" such as infotaxis. However, the gridworld navigation algorithm has an implicit allocentric representation, since movement can be in one of four allocentric directions (up, down, left, right), and wind direction is defined in these coordinates. Future studies might ask if an agent can learn the strategy without a known wind direction if it can only go left/right/forward/back/turn (in egocentric coordinates). In discussing possible algorithms, and the features of this one, it might be helpful to distinguish (1) those that rely only on egocentric computations (run and tumble), (2) those that rely on a single direction cue such as wind direction, (3) those that rely on allocentric representations of direction, and (4) those that rely on a full spatial map of the environment.

(2) Recovery strategy on losing the plume

The authors explore several recovery strategies upon losing the plume, including backtracking, circling, and learned strategies, finding that a learned strategy is optimal. As insects show a variety of recovery strategies that can depend on the model of locomotion, it would be interesting in the future to explore under which conditions various recovery strategies are optimal and whether they can predict the strategies of real animals in different environments.

(3) Is there a minimal representation of odor for efficient navigation?

The authors suggest that the number of olfactory states could potentially be reduced to reduce computational cost. They show that reducing the number of olfactory states to 1 dramatically reduces performance. In the future it would be interesting to identify optimal internal representations of odor for navigation and to compare these to those found in real olfactory systems. Does the optimal number of odor and void states depend on the spatial structure of the turbulence as explored in Figure 5?

---

## [Referee Report · Reviewer #2 (Public review)]

Summary:

The authors investigate the problem of olfactory search in turbulent environments using artificial agents trained using tabular Q-learning, a simple and interpretable reinforcement learning (RL) algorithm. The agents are trained solely on odor stimuli, without access to spatial information or prior knowledge about the odor plume's shape. This approach makes the emergent control strategy more biologically plausible for animals navigating exclusively using olfactory signals. The learned strategies show parallels to observed animal behaviors, such as upwind surging and crosswind casting. The approach generalizes well to different environments and effectively handles the intermittency of turbulent odors.

Strengths:

* The use of numerical simulations to generate realistic turbulent fluid dynamics sets this paper apart from studies that rely on idealized or static plumes.

* A key innovation is the introduction of a small set of interpretable olfactory states based on moving averages of odor intensity and sparsity, coupled with an adaptive temporal memory.

* The paper provides a thorough analysis of different recovery strategies when an agent loses the odor trail, offering insights into the trade-offs between various approaches.

* The authors provide a comprehensive performance analysis of their algorithm across a range of environments and recovery strategies, demonstrating the versatility of the approach.

* Finally, the authors list an interesting set of real-world experiments based on their findings, that might invite interest from experimentalists across multiple species.

Weaknesses:

* Using tabular Q-learning is both a strength and a limitation. It's simple and interpretable, making it easier to analyze the learned strategies, but the discrete action space seems somewhat unnatural. In real-world biological systems, actions (like movement) are continuous rather than discrete. Additionally, the ground-frame actions may not map naturally to how animals navigate odor plumes (e.g. insects often navigate based on their own egocentric frame).

---

## [Author Response]

The following is the authors’ response to the current reviews.

**Public Reviews:**

**Reviewer #1 (Public review):**
Overall I found the approach taken by the authors to be clear and convincing. It is striking that the conclusions are similar to those obtained in a recent study using a different computational approach (finite state controllers), and lends confidence to the conclusions about the existence of an optimal memory duration. There are a few questions that could be expanded on in future studies:(1) Spatial encoding requirementsThe manuscript contrasts the approach taken here (reinforcement learning in a gridworld) with strategies that involve a "spatial map" such as infotaxis. However, the gridworld navigation algorithm has an implicit allocentric representation, since movement can be in one of four allocentric directions (up, down, left, right), and wind direction is defined in these coordinates. Future studies might ask if an agent can learn the strategy without a known wind direction if it can only go left/right/forward/back/turn (in egocentric coordinates). In discussing possible algorithms, and the features of this one, it might be helpful to distinguish (1) those that rely only on egocentric computations (run and tumble), (2) those that rely on a single direction cue such as wind direction, (3) those that rely on allocentric representations of direction, and (4) those that rely on a full spatial map of the environment.

We agree that the question of what orientation skills are needed to implement an algorithm is interesting. We remark that our agents do not use allocentric directions in the sense of north, east, west and east relative to e.g. fixed landmarks in the environment. Instead, directions are defined *relative to the mean wind*, which is assumed fixed and known. (In our first answer to reviewers we used “north east south west relative to mean wind”, which may have caused confusion – but in the manuscript we only use upwind downwind and crosswind).

(2) Recovery strategy on losing the plumeThe authors explore several recovery strategies upon losing the plume, including backtracking, circling, and learned strategies, finding that a learned strategy is optimal. As insects show a variety of recovery strategies that can depend on the model of locomotion, it would be interesting in the future to explore under which conditions various recovery strategies are optimal and whether they can predict the strategies of real animals in different environments.

Agreed, it will be interesting to study systematically the emergence of distinct recovery strategies and compare to living organisms.

(3) Is there a minimal representation of odor for efficient navigation?The authors suggest that the number of olfactory states could potentially be reduced to reduce computational cost. They show that reducing the number of olfactory states to 1 dramatically reduces performance. In the future it would be interesting to identify optimal internal representations of odor for navigation and to compare these to those found in real olfactory systems. Does the optimal number of odor and void states depend on the spatial structure of the turbulence as explored in Figure 5?

We agree that minimal odor representations are an intriguing question. While tabular Q learning cannot derive optimal odor representations systematically, one could expand on the approach we have taken here and provide more comparisons. It will be interesting to follow this approach in a future study.

**Reviewer #2 (Public review):**
Summary:The authors investigate the problem of olfactory search in turbulent environments using artificial agents trained using tabular Q-learning, a simple and interpretable reinforcement learning (RL) algorithm. The agents are trained solely on odor stimuli, without access to spatial information or prior knowledge about the odor plume's shape. This approach makes the emergent control strategy more biologically plausible for animals navigating exclusively using olfactory signals. The learned strategies show parallels to observed animal behaviors, such as upwind surging and crosswind casting. The approach generalizes well to different environments and effectively handles the intermittency of turbulent odors.Strengths:* The use of numerical simulations to generate realistic turbulent fluid dynamics sets this paper apart from studies that rely on idealized or static plumes.* A key innovation is the introduction of a small set of interpretable olfactory states based on moving averages of odor intensity and sparsity, coupled with an adaptive temporal memory.* The paper provides a thorough analysis of different recovery strategies when an agent loses the odor trail, offering insights into the trade-offs between various approaches.* The authors provide a comprehensive performance analysis of their algorithm across a range of environments and recovery strategies, demonstrating the versatility of the approach.* Finally, the authors list an interesting set of real-world experiments based on their findings, that might invite interest from experimentalists across multiple species.Weaknesses:* Using tabular Q-learning is both a strength and a limitation. It's simple and interpretable, making it easier to analyze the learned strategies, but the discrete action space seems somewhat unnatural. In real-world biological systems, actions (like movement) are continuous rather than discrete. Additionally, the ground-frame actions may not map naturally to how animals navigate odor plumes (e.g. insects often navigate based on their own egocentric frame).

We agree with the reviewer, and will look forward to study this problem further to make it suitable for meaningful comparisons with animal behavior.

**Recommendations for the authors:**

**Reviewer #1 (Recommendations for the authors):**
The authors have addressed my major concerns and I support publication of this interesting manuscript. A couple of small suggestions:(1) In discussing performance in different environments (line 328-362) it might be easier to read if you referred to the environments by descriptive names rather than numbers.

Thank you for the suggestion, which we implemented

(2) Line 371: measurements of flow speed depend on antennae in insects. Insects can measure local speed and direct of flow using antennae, e.g. Bell and Kramer, 1979, Suver et al. 2019. Okubo et al. 2020,

Thank you for the references

(3) line 448: "Similarly, an odor detection elicits upwind surges that can last several seconds" maybe "Similarly, an odor detection elicits upwind surges that can outlast the odor by several seconds"?

Thank you for the suggestion

**Reviewer #2 (Recommendations for the authors):**
I commend the authors for their revisions in response to reviewer feedback.While I appreciate that the manuscript is now accompanied by code and data, I must note that the accompanying code-repository lacks proper instructions for use and is likely incomplete (e.g. where is the main function one should run to run your simulations? How should one train? How should one recreate the results? Which data files go where?).For examples of high-quality code-release, please see the documentation for these RL-for-neuroscience code repositories (from previously published papers):
https://github.com/ryzhang1/Inductive_bias

https://github.com/BruntonUWBio/plumetracknets
The accompanying data does provide snapshots from their turbulent plume simulations, which should be valuable for future research.

Thank you for the suggestions for how to improve clarity of the code. The way we designed the repository is to serve both the purpose of developing the code as well as sharing. This is because we are going to build up on this work to proceed further. Nothing is missing in the repository (we know it because it is what we actually use).

We do plan to create a more user-friendly version of the code, hopefully this will be ready in the next few months, but it wont be immediate as we are aiming to also integrate other aspects of the work we are currently doing in the Lab. The Brunton repository is very well organized, thanks for the pointer.

The following is the authors’ response to the original reviews.

**Reviewer #1 (Public review):**
Overall I found the approach taken by the authors to be clear and convincing. It is striking that the conclusions are similar to those obtained in a recent study using a different computational approach (finite state controllers), and lend confidence to the conclusions about the existence of an optimal memory duration. There are a few points or questions that could be addressed in greater detail in a revision:(1) Discussion of spatial encodingThe manuscript contrasts the approach taken here (reinforcement learning in a grid world) with strategies that involve a "spatial map" such as infotaxis. The authors note that their algorithm contains "no spatial information." However, I wonder if further degrees of spatial encoding might be delineated to better facilitate comparisons with biological navigation algorithms. For example, the gridworld navigation algorithm seems to have an implicit allocentric representation, since movement can be in one of four allocentric directions (up, down, left, right). I assume this is how the agent learns to move upwind in the absence of an explicit wind direction signal. However, not all biological organisms likely have this allocentric representation. Can the agent learn the strategy without wind direction if it can only go left/right/forward/back/turn (in egocentric coordinates)? In discussing possible algorithms, and the features of this one, it might be helpful to distinguish(1) those that rely only on egocentric computations (run and tumble),(2) those that rely on a single direction cue such as wind direction,(3) those that rely on allocentric representations of direction, and(4) those that rely on a full spatial map of the environment.

As Referee 1 points out, even if the algorithm does not require a map of space, the agent is still required to tell apart directions relative to the wind direction which is assumed known. Indeed, although in the manuscript we labeled actions allocentrically as “ up down left and right”, the source is always placed in the same location, hence “left” corresponds to upwind; “right” to downwind and “up” and “down” to crosswind right and left. Thus in fact directions are relative to the mean wind, which is therefore assumed known. We have better clarified the spatial encoding required to implement these strategies, and re-labeled the directions as upwind, downwind, crosswind-right and crosswind-left.

In reality, animals cannot measure the mean flow, but rather the local flow speed e.g. with antennas for insects, with whiskers for rodents and with the lateral line for marine organisms. Further work is needed to address how local flow measures enable navigation using Q learning.

(2) Recovery strategy on losing the plumeWhile the approach to encoding odor dynamics seems highly principled and reaches appealingly intuitive conclusions, the approach to modeling the recovery strategy seems to be more ad hoc. Early in the paper, the recovery strategy is defined to be path integration back to the point at which odor was lost, while later in the paper, the authors explore Brownian motion and a learned recovery based on multiple "void" states. Since the learned strategy works best, why not first consider learned strategies, and explore how lack of odor must be encoded or whether there is an optimal division of void states that leads to the best recovery strategies? Also, although the authors state that the learned recovery strategies resemble casting, only minimal data are shown to support this. A deeper statistical analysis of the learned recovery strategies would facilitate comparison to those observed in biology.

We thank Referee 1 for their remarks and suggestion to give the learned recovery a more prominent role and better characterize it. We agree that what is done in the void state is definitely key to turbulent navigation. In the revised manuscript, we have further substantiated the statistics of the learned recovery by repeating training 20 times and comparing the trajectories in the void (Figure 3 figure supplement 3, new Table 1). We believe however that starting with the heuristic recovery is clearer because it allows to introduce the concept of recovery more clearly. Indeed, the learned “recovery” is so flexible that it ends up mixing recovery (crosswind motion) to aspects of exploitation (surge): we defer a more in-depth analysis that disentangles these two aspects elsewhere. Also, we added a whole new comparison with other biologically inspired recoveries both in the native environment and for generalization (Figure 3 and 5).

(3) Is there a minimal representation of odor for efficient navigation?The authors suggest (line 280) that the number of olfactory states could potentially be reduced to reduce computational cost. This raises the question of whether there is a maximally efficient representation of odors and blanks sufficient for effective navigation. The authors choose to represent odor by 15 states that allow the agent to discriminate different spatial regimes of the stimulus, and later introduce additional void states that allow the agent to learn a recovery strategy. Can the number of states be reduced or does this lead to loss of performance? Does the optimal number of odor and void states depend on the spatial structure of the turbulence as explored in Figure 5?

We thank the referee for their comment. Q learning defines the olfactory states prior to training and does not allow a systematic optimization of odor representation for the task. We can however compare different definitions of the olfactory states, for example based on the same features but different discretizations. We added a comparison with a drastically reduced number of non-empty olfactory states to just 1, i.e. if the odor is above threshold at any time within the memory, the agent is in the non-void olfactory state, otherwise it is in the void state. This drastic reduction in the number of olfactory states results in less positional information and degrades performance (Figure 5 figure supplement 5).

The number of void states is already minimal: we chose 50 void states because this matches the time agents typically remain in the void (less than 50 void states results in no convergence and more than 50 introduces states that are rarely visited).

One may instead resort to deep Q-learning or to recurrent neural networks, which however do not provide answers as for what are the features or olfactory states that drive behavior (see discussion in manuscript and questions below).

**Reviewer #2 (Public review):**
Summary:The authors investigate the problem of olfactory search in turbulent environments using artificial agents trained using tabular Q-learning, a simple and interpretable reinforcement learning (RL) algorithm. The agents are trained solely on odor stimuli, without access to spatial information or prior knowledge about the odor plume's shape. This approach makes the emergent control strategy more biologically plausible for animals navigating exclusively using olfactory signals. The learned strategies show parallels to observed animal behaviors, such as upwind surging and crosswind casting. The approach generalizes well to different environments and effectively handles the intermittency of turbulent odors.Strengths:(1) The use of numerical simulations to generate realistic turbulent fluid dynamics sets this paper apart from studies that rely on idealized or static plumes.(2) A key innovation is the introduction of a small set of interpretable olfactory states based on moving averages of odor intensity and sparsity, coupled with an adaptive temporal memory.(3) The paper provides a thorough analysis of different recovery strategies when an agent loses the odor trail, offering insights into the trade-offs between various approaches.(4) The authors provide a comprehensive performance analysis of their algorithm across a range of environments and recovery strategies, demonstrating the versatility of the approach.(5) Finally, the authors list an interesting set of real-world experiments based on their findings, that might invite interest from experimentalists across multiple species.Weaknesses:(1) The inclusion of Brownian motion as a recovery strategy, seems odd since it doesn't closely match natural animal behavior, where circling (e.g. flies) or zigzagging (ants' "sector search") could have been more realistic.

We agree that Brownian motion may not be biologically plausible -- we used it as a simple benchmark. We clarified this point, and re-trained our algorithm with adaptive memory using circling and zigzaging (cast and surge) recoveries. The learned recovery outperforms all heuristic recoveries (Figure 3D, metrics G). Circling ranks second, and achieves these good results by further decreasing the probability of failure and paying slightly in speed. When tested in the non-native environments 2 to 6, the learned recovery performs best in environments 2, 5 and 6 i.e. from long range more relevant to flying insects; whereas circling generalizes best in odor rich environments 3 and 4, representative of closer range and close to the substrate (Figure 5B, metrics G). In the new environments, similar to the native environment, circling favors convergence (Figure 5B, metrics f^+^) over speed (Figure 5B, metrics g^+^ and τ_min_/τ), which is particularly deleterious at large distance.

(2) Using tabular Q-learning is both a strength and a limitation. It's simple and interpretable, making it easier to analyze the learned strategies, but the discrete action space seems somewhat unnatural. In real-world biological systems, actions (like movement) are continuous rather than discrete. Additionally, the ground-frame actions may not map naturally to how animals navigate odor plumes (e.g. insects often navigate based on their own egocentric frame).

We agree with the reviewer that animal locomotion does not look like a series of discrete displacements on a checkerboard. However, to overcome this limitation, one has to first focus on a specific system to define actions in a way that best adheres to a species’ motor controls. Moreover, these actions are likely continuous, which makes reinforcement learning notoriously more complex. While we agree that more realistic models are definitely needed for a comparison with real systems, this remains outside the scope of the current work. We have added a remark to clarify this limitation.

(3) The lack of accompanying code is a major drawback since nowadays open access to data and code is becoming a standard in computational research. Given that the turbulent fluid simulation is a key element that differentiates this paper, the absence of simulation and analysis code limits the study's reproducibility.

We have published the code and the datasets at

- code: https://github.com/Akatsuki96/qNav

- datasets: https://zenodo.org/records/14655992

**Recommendations for the authors:**

**Reviewer #1 (Recommendations for the authors):**
(1) Line 59-69: In comparing the results here to other approaches (especially the Verano and Singh papers), it would also be helpful to clarify which of these include an explicit representation of the wind direction. My understanding is that both the Singh and Verano approaches include an explicit representation of wind direction. In Singh wind direction is one of the observations that inputs to the agent, while in Verano, the actions are defined relative to the wind direction. In the current paper, my understanding is that there is no explicitly defined wind direction, but because movement directions are encoded allocentrically, the agent is able to learn the upwind direction from the structure of the plume- is this correct? I think this information would be helpful to spell out and also to address whether an agent without any allocentric direction sense can learn the task.

Thank you for the comment. In our algorithm the directions are defined relative to the mean wind, which is assumed known, as in Verano et al. As far as we understand, Singh et al provide the instantaneous, egocentric wind velocities as part of the input.

(1) Line 105: "several properties of odor stimuli depend on the distance from the source" might cite Boie...Victor 2018, Ackles...Schaefer, 2021, Nag...van Breugel 2024.

Thank you for the suggestions - we have added these references

(2) Line 130: "we first define a finite set of olfactory states" might be helpful to the reader to state what you chose in this paragraph rather than further down.

We have slightly modified the incipit of the paragraph. We first declare we are setting out to craft the olfactory states, then define the challenges, finally we define the olfactory states.

(3) Line 267: "Note that the learned recovery strategy resembles casting behavior observed in flying insects" Might note that insects seem to deploy a range of recovery strategies depending on locomotor mode and environment. For example, flying flies circle and sink when odor is lost in windless environments (Stupski and van Breugel 2024).

Thank you for your comment. We have included the reference and we now added comparisons to results using circling and cast & surge recovery strategies.

(4) Line 289: "from positions beyond the source, the learned strategy is unable to recover the plume as it mostly casts sideways, with little to no downwind action" This is curious as many insects show a downwind bias in the absence of odor that helps them locate the plumes in the first place (e.g. Wolf and Wehner, 2000, Alvarez-Salvado et al. 2018). Is it possible that the agent could learn a downwind bias in the absence of odor if given larger environments or a longer time to learn?

The reviewer is absolutely correct – Downwind motion is not observed in the recovery simply because the agent rarely overshoots the source. Hence overall optimization for that condition is washed out by the statistics. We believe downwind motion will emerge if an agent needs to avoid overshooting the source – we do not have conclusive results yet but are planning to introduce such flexibility in a further work. We added this remark and refs.

(5) Line 377-391: testing these ideas in living systems. Interestingly, Kathman..Nagel 2024 (bioRxiv) shows exactly the property predicted here and in Verano in fruit flies- an odor memory that outlasts the stimulus by a duration of several seconds, appropriate for filling in "blanks." Relatedly, Alvarez-Salvado et al. 2018 showed that fly upwind running reflected a temporal integration of odor information over ~10s, sufficient to avoid responding to blanks as loss of odor.

Indeed, we believe this is the most direct connection between algorithms and experiments. We are excited to discuss with our colleagues and pursue a more direct comparison with animal behavior. We were aware of the references and forgot to cite them, thank you for your careful reading of our work !

**Reviewer #2 (Recommendations for the authors):**
Suggestions(1) The paper does not clearly specify which type of animals (e.g., flying insects, terrestrial mammals) the model is meant to approximate or not approximate. The authors should consider clarifying how these simulations are suited to be a general model across varied olfactory navigators. Further, it isn't clear how low/high the intermittency studied in this model is compared to what different animals actually encounter. (Minor: The Figure 4 occupancy circles visualization could be simplified).

Environment 1 represents the lower layers of a moderately turbulent boundary layer. Search occurs on a horizontal plane ~half meter from the ground. The agent is trained at distances of about 10 meters and also tested on longer distances ~ 17 meters (environment 6), lower heights ~1cm from the ground (environments 3-4), lower Reynolds number (environment 5) and higher threshold of detection (environment 2 and 4). Thus Environments 1,2,5 and 6 are representative of conditions encountered by flying organisms (or pelagic in water), and Environments 3 and 4 of searches near the substrate, potentially involved in terrestrial navigation (benthic in water). Even near the substrate, we use odor dispersed in the fluid, and not odor attached to the substrate (relevant to trail tracking).

Also note that we pick Schmidt number Sc = 1 and this is appropriate for odors in air but not in water. However, we expect a weak dependence on the Schmidt number as the Batchelor and Kolmogorov scales are below the size of the source and we are interested in the large scale statistics Falkovich et al., 2001; Celani et al., 2014; Duplat et al., 2010.

Intermittency contours are shown in Fig 1C, they are highest along the centerline, and decay away from the centerline, so that even within the plume detecting odor is relatively rare. Only a thin region near the centerline has intermittency larger than 66%; the outer and most critical bin of the plume has intermittency under 33%; in the furthest point on the centerline intermittency is <10%. For reference, experimental values in the atmospheric boundary layer report intermittency 25% to 20% at 2 to 15m from the source along the centerline (Murlis and Jones, 1981).

We have more clearly labeled the contours in Fig 1C and added these remarks.

We included these remarks and added a whole table with matching to real conditions within the different environments.

(2) Could some biological examples and references be added to support that backtracking is a biologically plausible mechanism?

Backtracking was observed e.g. in ants displaced in unfamiliar environments (Wystrach et al, P Roy Soc B, 280, 2013), in tsetse flies executing reverse turns uncorrelated to wind, which bring them back towards the location where they last detected odor (Torr, Phys Entom, 13, 1988, Gibson & Brady Phys Entom 10, 1985) and in coackroaches upon loss of contact with the plume (Willis et al, J. Exp. Biol. 211, 2008). It is also used in computational models of olfactory navigation (Park et al, Plos Comput Biol, 12:e1004682, 2016).

(3) Hand-crafted features can be both a strength and a limitation. On the one hand, they offer interpretability, which is crucial when trying to model biological systems. On the other hand, they may limit the generality of the model. A more thorough discussion of this paper's limitations should address this.(4) The authors mention the possibility of feature engineering or using recurrent neural networks, but a more concrete discussion of these alternatives and their potential advantages/disadvantages would be beneficial. It should be noted that the hand-engineered features in this manuscript are quite similar to what the model of Singh et al suggests emerges in their trained RNNs.

Merged answer to points 3 and 4.

We agree with the reviewer that hand-crafted features are both a strength and a limitation in terms of performance and generality. This was a deliberate choice aimed at stripping the algorithm bare of implicit components, both in terms of features and in terms of memory. Even with these simple features, our model performs well in navigating across different signals, consistent with our previous results showing that these features are a “good” surrogate for positional information.

To search for the most effective temporal features, one may consider a more systematic hand crafting, scaling up our approach. In this case one would first define many features of the odor trace; rank groups of features for their accuracy in regression against distance; train Q learning with the most promising group of features and rank again. Note however that this approach will be cumbersome because multiple factors will have to be systematically varied: the regression algorithm; the discretization of the features and the memory.

Alternatively, to eliminate hand crafting altogether and seek better performance or generalization, one may consider replacing these hand-crafted features and the tabular Q-learning approach with recurrent neural networks or with finite state controllers. On the flip side, neither of these algorithms will directly provide the most effective features or the best memory, because these properties are hidden within the parameters that are optimized for. So extra work is needed to interrogate the algorithms and extract these information. For example, in Singh et al, the principal components of the hidden states in trained agents correlate with head direction, odor concentration and time since last odor encounter. More work is needed to move beyond correlations and establish more systematically what are the features that drive behavior in the RNN.

We have added these points to the discussion.

(5) Minor: the title of the paper doesn't immediately signal its focus on recovery strategies and their interplay with memory in the context of olfactory navigation. Given the many other papers using a similar RL approach, this might help the authors position this paper better.

We agree with the referee and have modified the title to reflect this.

(6) Minor: L 331: "because turbulent odor plumes constantly switch on and off" -- the signal received rather than the plume itself is switching on and off.

Thank you for the suggestion, we implemented it.